# Plant pathogens convergently evolved to counteract redundant nodes of an NLR immune receptor network

**Lida Derevnina**[1], **Mauricio P. Contreras**[1], **Hiroaki Adachi**[1¤], **Jessica Upson**[1], **Angel Vergara Cruces**[1,2], **Rongrong Xie**[1,3], **Jan Sklenar**[1], **Frank L. H. Menke**[1], **Sam T. Mugford**[4], **Dan MacLean**[1], **Wenbo Ma**[1,5], **Saskia A. Hogenhout**[4], **Aska Goverse**[6], **Abbas Maqbool**[1], **Chih-Hang Wu**[1,7]*, **Sophien Kamoun**[1]*

1 The Sainsbury Laboratory, University of East Anglia, Norwich, United Kingdom, 2 Department of Biology, Swiss Federal Institute of Technology (ETH), Zürich, Switzerland, 3 Joint Center for Single Cell Biology, School of Agriculture and Biology, Shanghai, Jiao Tong University, Shanghai, China, 4 Department of Crop Genetics, John Innes Centre, Norwich, United Kingdom, 5 Department of Plant Pathology and Microbiology, University of California, Riverside, California, United States of America, 6 Laboratory of Nematology, Wageningen University and Research, Wageningen, the Netherlands, 7 Institute of Plant and Microbial Biology, Academia Sinica, Taipei, Taiwan

¤ Current address: Graduate School of Biological Sciences, Nara Institute of Science and Technology, Ikoma, Japan
* wuchh@gate.sinica.edu.tw (C-HW); Sophien.Kamoun@tsl.ac.uk (SK)

**Data Availability Statement:** All relevant data are within the paper and its Supporting Information files. The MS proteomics data have been deposited to the ProteomeXchange Consortium via the

## Abstract

In plants, nucleotide-binding domain and leucine-rich repeat (NLR)-containing proteins can form receptor networks to confer hypersensitive cell death and innate immunity. One class of NLRs, known as NLR required for cell death (NRCs), are central nodes in a complex network that protects against multiple pathogens and comprises up to half of the NLRome of solanaceous plants. Given the prevalence of this NLR network, we hypothesised that pathogens convergently evolved to secrete effectors that target NRC activities. To test this, we screened a library of 165 bacterial, oomycete, nematode, and aphid effectors for their capacity to suppress the cell death response triggered by the NRC-dependent disease resistance proteins Prf and Rpi-blb2. Among 5 of the identified suppressors, 1 cyst nematode protein and 1 oomycete protein suppress the activity of autoimmune mutants of NRC2 and NRC3, but not NRC4, indicating that they specifically counteract a subset of NRC proteins independently of their sensor NLR partners. Whereas the cyst nematode effector SPRYSEC15 binds the nucleotide-binding domain of NRC2 and NRC3, the oomycete effector AVRcap1b suppresses the response of these NRCs via the membrane trafficking-associated protein NbTOL9a (Target of Myb 1-like protein 9a). We conclude that plant pathogens have evolved to counteract central nodes of the NRC immune receptor network through different mechanisms. Coevolution with pathogen effectors may have driven NRC diversification into functionally redundant nodes in a massively expanded NLR network.

PRIDE partner repository with the dataset identifier PXD023178 and 10.6019/PXD023178.

**Funding:** This work was funded by the Gatsby Charitable Foundation (Core grant to TSL) and Biotechnology and Biological Sciences Research Council (BBSRC, UK) BB/P012574 and BB/V002937/1. S.K. also receives funding from the European Research Council (ERC NGRB and BLASTOFF projects). L.D. was funded by a Marie Sklodowska-Curie actions fellowship (BoostR), H. A. was funded by the Japan Society for the Promotion of Sciences Postdoctoral fellowship, A. V.C. was funded by a British Society for Plant Pathology summer bursary, J.U. was funded by the Gatsby Charitable Foundation PhD studentship. The funders had no role in study design, data collection and analysis, decision to publish, or preparation of the manuscript.

**Competing interests:** I have read the journal's policy and the authors of this manuscript have the following competing interests: S.K. receives funding from industry on NLR biology.

**Abbreviations:** AVR, avirulence; CaMV, cauliflower mosaic virus; CC, coiled-coil; coIP, co-immunoprecipitation; CP, coat protein; ESCRT, endosomal sorting complex required for transport; ETI, effector-triggered immunity; EV, empty vector; HR, hypersensitive response; HRP, horse radish peroxidase; LC–MS/MS, liquid chromatography–tandem mass spectrometry; LRR, leucine-rich repeat; MAPKK, mitogen-activated protein kinase kinase; MHD, methionine–histidine–aspartate; NB-ARC, nucleotide-binding domain shared with APAF-1, various R-proteins, and CED-4; NLR, nucleotide-binding domain and leucine-rich repeat; NRC, NLR required for cell death; NTI, NLR-triggered immunity; PAMP, pathogen-associated molecular pattern; PRR, pattern recognition receptor; PTI, PRR-triggered immunity; PVX, *Potato virus X*; ROQ1, Recognition of XopQ 1; RPP1, Recognition of Peronospora parasitica 1; TIR, Toll/interleukin-1 receptor; TOL, Target of Myb 1-like protein; TRV, *Tobacco rattle virus*; VIGS, virus-induced gene silencing; Y2H, yeast two-hybrid; ZAR1, HopZ-activated resistance 1.

## Introduction

Our view of the pathogenicity mechanisms of plant pathogens and pests has significantly broadened over the years. Parasites as diverse as bacteria, oomycetes, nematodes, and aphids turned out to be much more sophisticated manipulators of their host plants than initially anticipated. Indeed, it is now well established that these parasites secrete an arsenal of proteins, termed effectors, which modulate plant responses, such as innate immunity, to enable host infection and colonisation. As a consequence, deciphering the biochemical activities of effectors to understand how parasites successfully colonise and reproduce has become a major conceptual paradigm in the field of molecular plant pathology [1,2]. In fact, effectors have emerged as molecular probes that can be utilised to unravel novel components and processes of the host immune system [2].

Most effectors studied to date suppress immune pathways induced by pathogen-associated molecular patterns (PAMPs). This so-called PAMP or pattern recognition receptor (PRR)-triggered immunity (PTI) response is mediated by cell surface PRRs [3,4] (S1A Fig). A subset of effectors, however, have avirulence (AVR) activity and inadvertently activate intracellular immune receptors of the nucleotide-binding, leucine-rich repeat (NLR) class of proteins, a response known as effector- or NLR-triggered immunity (ETI/NTI) [5–7]. In plants, recognition of AVR effectors by NLRs can occur either directly or indirectly following various mechanistic models [8,9]. NTI is usually accompanied by a localised form of programmed cell death known as the hypersensitive response (HR) that hinders disease progression [6,7,10,11]. NLR-mediated immunity activated by AVR effectors can also be suppressed by other effectors [12]. However, in sharp contrast to the widely studied PTI-suppressing effectors, the mechanisms by which effectors suppress NLR responses remain poorly understood [3,13,14]. Therefore, understanding how effectors suppress NLR functions should provide important insights into the black box of how these immune receptors activate cell death and innate immunity, one of the major unsolved questions in the field of plant pathology [15].

NLRs are multidomain proteins with a nucleotide-binding domain shared with APAF-1, various R-proteins, and CED-4 (NB-ARC) and at least 1 additional domain [16]. The C-terminus of plant NLRs is generally a leucine-rich repeat (LRR) domain, but they can be sorted into phylogenetic subgroups with distinct N-terminal domains [16–18]. In angiosperms, NLRs form several major phylogenetic subgroups, including TIR-NLRs with an N-terminal Toll/interleukin-1 receptor (TIR) domain, CC-NLRs, with the Rx-type coiled-coil (CC) domain, $CC_R$-NLRs with the RPW8-type CC ($CC_R$) domain, and the $CC_{G10}$-NLR subclade with a distinct type of CC ($CC_{A\ or\ CC_{G10}}$) domain. Of these, CC-NLRs are the most common type, forming the largest group of NLRs in angiosperms [16,18,19]. The TIR- and CC-type N-terminal domains are involved in downstream immune signalling, oligomerization, and cell death execution [20]. The central NB-ARC domain exhibits ATP binding and hydrolysis activities and functions as a molecular switch that determines the NLR inactive/active status [21]. Finally, the LRR domain can mediate effector perception and often engages in intramolecular interactions [8,21].

Plant genomes may encode anywhere between 50 and approximately 1,000 NLRs [17,22]. Some of these NLRs are functional singletons operating as single biochemical units, while others function in pairs or in more complex receptor networks [23]. Paired and networked NLRs consist of functionally specialised sensor NLRs that detect pathogen effectors and helper NLRs that translate this effector recognition into hypersensitive cell death and resistance. In the Solanaceae, a major phylogenetic clade of NLRs forms a complex immunoreceptor network in which multiple helper CC-NLRs, known as NLR required for cell death (NRCs), are necessary for a large number of sensor NLRs [24]. These sensor NLRs, encoded by R gene loci, confer

resistance against diverse pathogens, such as viruses, bacteria, oomycetes, nematodes, and insects [24] (S1B Fig). Together, NRCs and their NLR partners form the NRC superclade, a well-supported phylogenetic cluster divided into an NRC helper clade and 2 larger clades that include all known NRC-dependent sensor NLRs [24,25]. The NRC superclade emerged over 100 million years ago (Mya) from an NLR pair that expanded to make up a significant fraction of the NLRome of asterid plant species [24]. The current model is that NRCs form redundant central nodes in this massively diversified bow tie network, with different NRCs exhibiting some specificity towards their sensor NLR partners. For example, whereas the bacterial resistance protein Prf requires NRC2 and NRC3 in a genetically redundant manner, the potato blight resistance protein Rpi-blb2 is dependent only on NRC4 [24,26] (S1B Fig).

Until very recently, the molecular mechanisms that underpin plant NLR activation and the subsequent execution of NTI pathways and hypersensitive cell death have remained unknown. The recent elucidation of the CC-NLR ZAR1 (HopZ-activated resistance 1) and the TIR-NLRs ROQ1 (Recognition of XopQ 1) and RPP1 (Recognition of *Peronospora parasitica* 1) structures have resulted in a new conceptual framework [27–30]. Activated NLRs oligomerize into a wheel-like multimeric complex known as the resistosome. In the case of ZAR1, activation induces conformational changes in the NB-ARC domain resulting in ADP release followed by dATP/ATP binding and oligomerization of ZAR1 and its partner host proteins into the pentameric resistosome structure [28,31]. ZAR1 resistosomes expose a funnel-shaped structure formed by the N-terminal α1 helices, which was proposed to perturb plasma membrane integrity to trigger hypersensitive cell death [28,29]. This α1 helix is defined by a molecular signature, the MADA motif, which is present in approximately 20% of angiosperm CC-NLRs, including NRCs [25]. This implies that the biochemical "death switch" mechanism of the ZAR1 resistosome probably applies to NRCs and other MADA motif containing NLRs across diverse plant taxa.

The NRC superclade is massively expanded in solanaceous plants, where it comprises up to half of the NLRome in some species [24]. Given this prevalence, we hypothesised that pathogens convergently evolved to secrete effectors that target the NRC network. In this study, we screened collections of effector proteins from 6 major parasites of solanaceous plants, i.e., the bacterium *Pseudomonas syringae*, the oomycete *Phytophthora infestans*, the cyst nematodes *Globodera rostochiensis* and *Globodera pallida*, and the aphids *Myzus persicae* and *Acrythosiphon pisum*, for their capacity to suppress the hypersensitive cell death triggered by the NRC-dependent sensor NLRs Prf and Rpi-blb2. These screens revealed 5 effectors as suppressors of components of the NRC network. Among these, the oomycete protein PITG-16705 (PBHR_012, henceforth, AVRcap1b) [32] and the cyst nematode protein SPRYSEC15 (where the term SPRYSEC will be referred to as SS from here on in) stood out for being able to specifically suppress the response triggered by autoimmune mutants of NRC2 and NRC3, but not NRC4, indicating that they are able to counteract a subset of NRC helper proteins independently of their sensor NLR partners. Further studies revealed that AVRcap1b and SS15 suppress NRC2 and NRC3 through different mechanisms. While AVRcap1b suppression of NRC2 and NRC3 mediated immunity is dependent on the membrane trafficking-associated protein NbTOL9a (Target of Myb 1-like protein 9a), SS15 directly binds the NB-ARC domain of NRC2 and NRC3 to perturb their function. We conclude that evolutionarily divergent plant pathogens have convergently evolved distinct molecular strategies to counteract central nodes of the NRC immune receptor network. Coevolution with pathogen effectors may, therefore, underpin the diversification of NRCs as functionally redundant nodes in a massively expanded NLR network.

## Results

### *In planta* screens reveal pathogen effectors that suppress NRC-mediated hypersensitive cell death

To determine the extent to which pathogens have evolved to target the NRC network, we screened candidate effectors from 6 diverse pathogen species for their ability to suppress the hypersensitive cell death triggered by the disease resistance proteins Prf (responds to Pto/AvrPto; NRC2/3 dependent) and Rpi-blb2 (responds to AVRblb2; NRC4 dependent) [24,26]. The effectors and empty vector (EV, control) were transiently expressed with Pto/AvrPto or Rpi-blb2/AVRblb2 in leaves of the model experimental plant *Nicotiana benthamiana* and assessed for the absence of cell death, which is indicative of a suppression phenotype (Fig 1A). Our screen included 165 effector candidates from oomycetes (*P. infestans*, 63), bacteria (*P. syringae*, 26), cyst nematodes (*G. rostochiensis*, 23 and *G. pallida*, 3), and aphids (*M. persicae*, 47 and *A. pisum*, 3). The panel of effectors used in the screen was comprised of candidates with previously reported involvement in immune suppression and effectors that were readily available in the lab. Among the 165 effectors tested, 2 effectors (AVRcap1b and SS15) suppressed Prf-mediated cell death, and 3 effectors (PITG-15278, SS10, and SS34) suppressed Rpi-blb2–mediated cell death (Figs 1B–1D and S2, S1–S4 Data). AVRcap1b and PITG-15278 are RXLR-WY/LWY domain containing effectors from *P. infestans* [33,34]. SS10, SS15, and SS34 are SPRY domain containing effectors from *G. rostochiensis* [35–37] (Fig 2A, S1 Table). Interestingly, HopAB2 (also known as AvrPtoB), which is well known to suppress Prf-mediated cell death [38], was not a robust suppressor in our assays, and only suppressed Prf-mediated cell death in older leaves of *N. benthamiana* (S3 Fig), leaf age as defined by [39].

### AVRcap1b and SS15 effectors suppress the cell death triggered by autoimmune mutants of NRC2 and NRC3 but not NRC4

To ascertain whether the identified effectors suppress the activity of the sensor NLRs (Prf or Rpi-blb2) or the underlying helper NLRs (NRC2, NRC3, and NRC4), we generated constitutively active (autoimmune) NRCs by mutating the methionine–histidine–aspartate (MHD) motif resulting in the NRC autoactive variants NRC2$^{H480R}$, NRC3$^{D480V}$, and NRC4$^{D478V}$ [40] (Fig 2B). When transiently expressed in *N. benthamiana*, all 3 NRC mutants caused cell death in the absence of a pathogen AVR effector (Fig 2C). Next, we coexpressed the suppressors AVRcap1b, SS15, PITG-15278, SS10, and SS34, along with the EV control, with NRC2$^{H480R}$, NRC3$^{D480V}$, and NRC4$^{D478V}$ in *N. benthamiana* leaves (S4 Fig, S4 Data). PITG-15278, SS10, and SS34 did not suppress any of the autoactive NRCs, indicating that they target the sensor NLR Rpi-blb2 or the interaction between Rpi-blb2 and NRC4 (S4 Fig). Interestingly, both AVRcap1b and SS15 suppressed the cell death induced by autoimmune NRC2 and NRC3, therefore recapitulating their specific suppression of the sensor NLRs Prf (Figs 2C and 2D and S5, S5 Data). In contrast, AVRcap1b and SS15 did not affect the autoactivity of NRC4 (Figs 2C and 2D and S5). These results indicate that the suppression activity of AVRcap1b and SS15 is independent of the sensor NLR Prf and is specific to NRC2 and NRC3. In light of these results, we focused on AVRcap1b and SS15 as they are likely acting on the NRC2 and NRC3 helper nodes of the NRC network and thus hold more potential to shed light on the molecular mechanisms underpinning helper NLR activation and downstream signalling.

### AVRcap1b and SS15 effectors suppress multiple disease resistance proteins (sensor NLRs) that signal through NRC2 and NRC3

To further challenge our finding that AVRcap1b and SS15 target NRC2 and NRC3 but not NRC4, we screened the 2 effectors for suppression of 6 disease resistance sensor NLRs (SW5b,

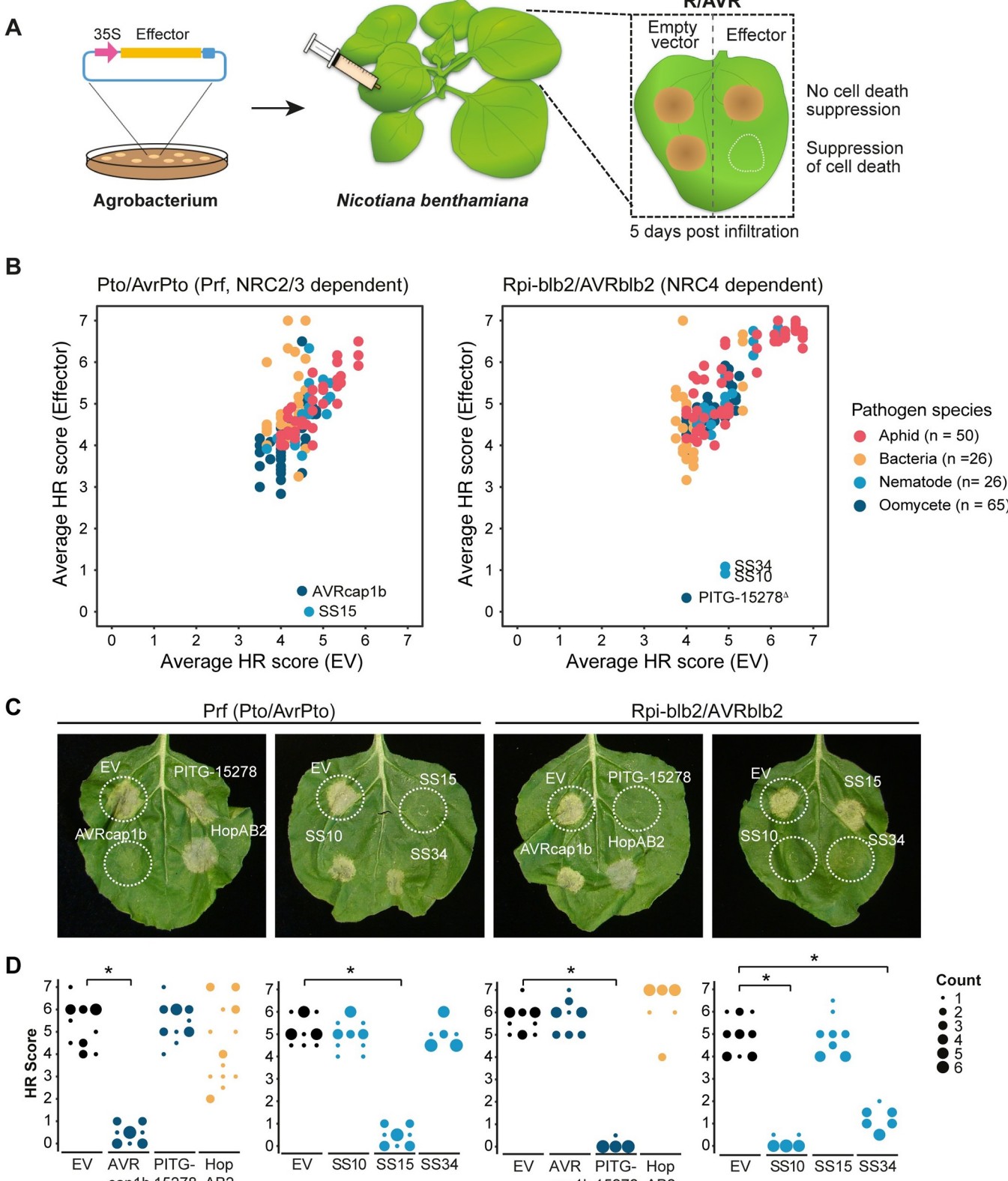

**Fig 1. Five effectors suppress Prf- (NRC2/3 dependent) or Rpi-blb2– (NRC4 dependent) mediated cell death.** (A) A schematic representation of cell death assay strategy used. Effectors and EV were transformed into *Agrobacterium tumefaciens* and transiently coexpressed in *N. benthamiana* plants with either Pto/AvrPto or Rpi-blb2/AVRblb2. EV was used as a negative control. HR was scored based on a modified 0–7 scale (Segretin and colleagues) between 5–7 days post-infiltration. (B) A scatterplot of the average HR score of EV versus the average HR score of each tested effector (*n* = 165); Pto/AvrPto (left panel) Rpi-blb2/

AVRblb2 (right panel). Effectors that have suppression activity are represented as outliers within the plot. Results are based on 6 technical replicates (S1–S3 Data). △Two alleles of PITG-15278 (*P. infestans* strain T30-4 and 17777) suppressed Rpiblb2-mediated cell death. PITG-15278 T30-4 is used as a representative in subsequent experiments. (C) Representative *N. benthamiana* leaves infiltrated with appropriate constructs photographed 5–7 days after infiltration. The R/AVR pair tested, Prf (Pto/AvrPto) or Rpi-blb2/AVRblb2, are labelled above the leaf panel and effectors labelled on leaf image. (D) HR was scored 5 days post-agroinfiltration (S4 Data). The results are presented as a dot plot, where the size of each dot is proportional to the number of samples with the same score (count) within each biological replicate. The experiment was independently repeated 3 times with 6 technical replicates. The columns for either EV or for each individual effector correspond to results from different biological replicates. Dot colours represent pathogen species as indicated in (B). Statistical tests were implemented using the besthr R library [41]. We performed bootstrap resampling tests using a lower significance cutoff of 0.025 and an upper cutoff of 0.975. Mean ranks of test samples falling outside of these cutoffs in the control samples bootstrap population were considered significant. Significant differences between the conditions are indicated with an asterisk (*). Details of statistical analysis are presented in S2 Fig. AVR, avirulence; EV, empty vector; HR, hypersensitive response.

Gpa2, R1, Rx, Bs2, and R8) that were previously assigned to the NRC network [24,25]. We transiently coexpressed AVRcap1b and SS15 with these NRC-dependent sensor NLRs in addition to the previously tested Prf and Rpi-blb2 in *N. benthamiana* leaves using agroinfiltration. Of these, 5 NLRs were coexpressed with their cognate AVR effectors, whereas we used a constitutively active version of SW5b (SW5b$^{D857V}$). We also included Rpi-vnt1 as an NRC-independent NLR protein negative control [24]. Interestingly, while SW5b was previously shown to signal through NRC2, NRC3, and NRC4 in a redundant manner [24], SW5b$^{D857V}$ signalled through NRC2 and NRC3 only. These experiments revealed that AVRcap1b and SS15 suppress SW5b$^{D857V}$ and Gpa2 in addition to Prf, but none of the other 5 NLRs (Rpi-blb2, R8, R1, Rx, and Bs2). Our findings do, however, contrast to a previous study that showed that SS15 was able to partially suppress Rx-mediated cell death [35]. Additionally, neither AVRcap1b nor SS15 suppressed cell death mediated by the NRC-independent NLR Rpi-vnt1 (Figs 3 and S6 and S6 Data). These results are consistent with the model that AVRcap1b and SS15 suppress NRC2 and NRC3 but not NRC4, given that, similar to Prf, both SW5b$^{D857V}$ and Gpa2 are dependent on NRC2 and NRC3, whereas the other tested NLRs signal through NRC4 specifically or redundantly with NRC2 and NRC3 [24,25].

## AVRcap1b and SS15 suppress Rx-mediated cell death only in the absence of NRC4

The NRC-dependent sensor NLR Rx, which confers extreme resistance to *Potato virus X* (PVX) by recognising viral coat protein (CP), is dependent on NRC2, NRC3, and NRC4 in a genetically redundant manner [42–45]. The genetic redundancy of NRCs may explain why the Rx-mediated cell death is not suppressed by AVRcap1b and SS15 in *N. benthamiana* (Fig 3). Since AVRcap1b and SS15 suppress the cell death activity of NRC2 and NRC3 but not NRC4, we reasoned that these 2 effectors should be able to suppress Rx-mediated cell death in the absence of NRC4. To challenge this hypothesis, we knocked down NRCs in Rx-transgenic *N. benthamiana* plants using *Tobacco rattle virus* (TRV)-induced gene silencing, either individually (TRV:*NRC4*) or in combination (TRV:*NRC2/3*) (Fig 4A). Three weeks after inoculation with TRV, we coexpressed AVRcap1b or SS15 with CP in *NRC*-silenced leaves. These experiments showed that both AVRcap1b and SS15 suppress Rx-mediated cell death in *NRC4*-silenced leaves, but not in *NRC2/3*-silenced leaves nor in the TRV:EV control (Figs 4B–4E and S7 and S7 Data). These results further validate our earlier finding that AVRcap1b and SS15 can specifically suppress the activities of NRC2 and NRC3 but not NRC4.

## AVRcap1b and SS15 compromise Rx-mediated extreme resistance to *Potato virus X* in an NRC2- and NRC3- but not NRC4-dependent manner

We previously showed that cosilencing of *NRC2*, *NRC3*, and *NRC4* not only compromised Rx-mediated hypersensitive cell death but also abolished extreme resistance to PVX, leading to

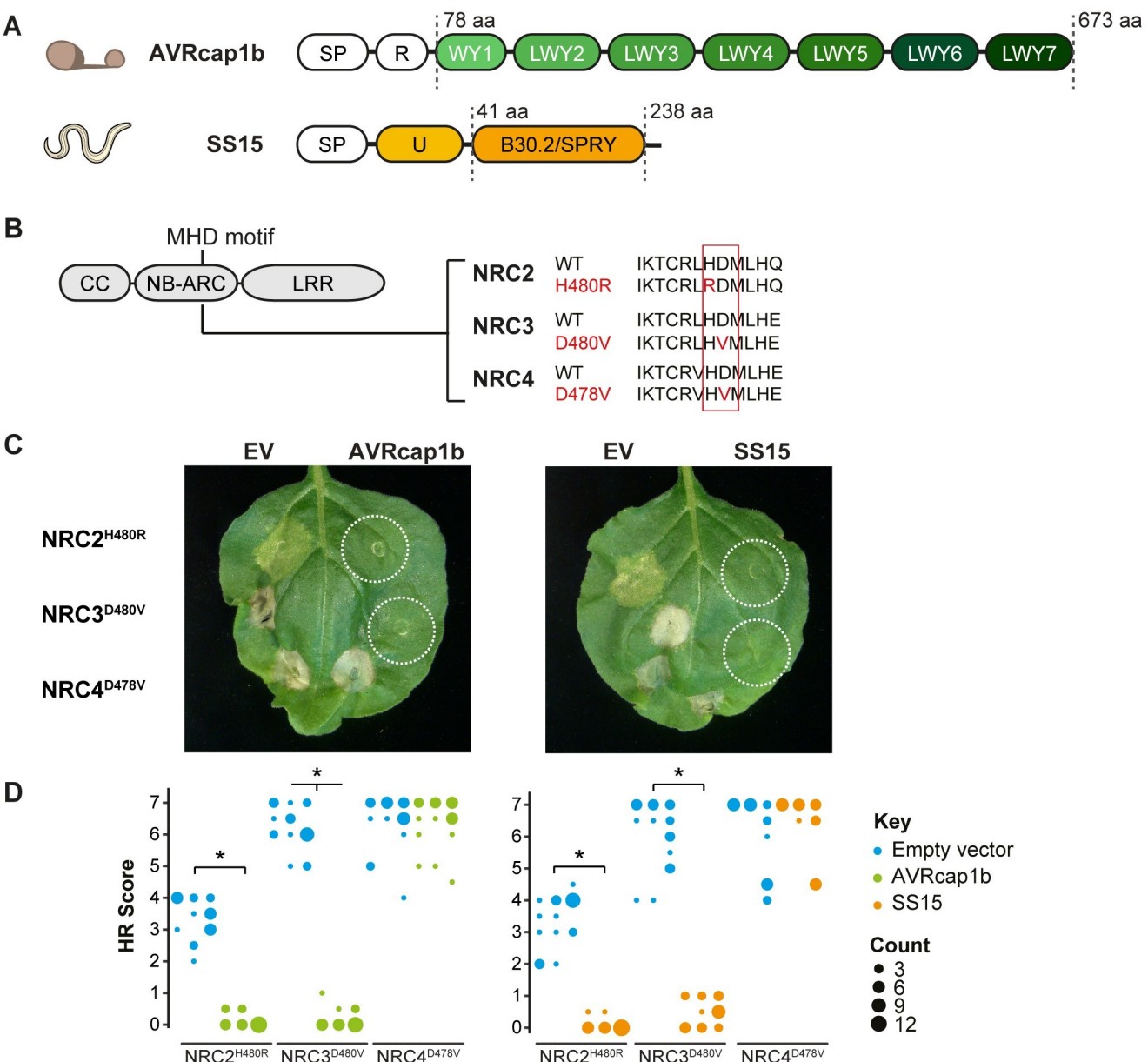

**Fig 2. AVRcap1b and SS15 suppress cell death activity mediated by autoactive NRC2$^{H480R}$ and NRC3$^{D480V}$ but not NRC4$^{D478V}$.** (A) Domain organisation of the RXLR-WY/LWY domain containing *P. infestans* effector AVRcap1b and the B30.2/SPRY (IPR001870) domain containing *G. rostochiensis* effector SS15, where SP = signal peptide, R = RxLR motif, U = uncharacterized domain. (B) Schematic representation of NRC2, NRC3, and NRC4 and the mutated sites in the MHD motif. Substituted residues are shown in red in the multiple sequence alignment. (C) Representative *N. benthamiana* leaves showing HR after coexpression of EV, AVRcap1b, or SS15, indicated above leaf panels, with autoimmune NRC2$^{H480R}$, NRC3$^{D480V}$, and NRC4$^{D478V}$ mutants. Plants were photographed 5 days after agroinfiltration. (D) HR was scored 5 days post-agroinfiltration (S5 Data). The results are presented as a dot plot, where the size of each dot is proportional to the number of samples with the same score (count) within each biological replicate. The experiment was independently repeated 3 times each with 6 technical replicates. The columns of each tested condition (labelled on the bottom of the plot) correspond to results from different biological replicates. Statistical tests were implemented using the besthr R library [41]. We performed bootstrap resampling tests using a lower significance cutoff of 0.025 and an upper cutoff of 0.975. Mean ranks of test samples falling outside of these cutoffs in the control samples bootstrap population were considered significant. Significant differences between the conditions are indicated with an asterisk (*). Details of statistical analysis are presented in S4 Fig. CC, coiled-coil; EV, empty vector; HR, hypersensitive response; LRR, leucine-rich repeat; MHD, methionine–histidine–aspartate; NB-ARC, nucleotide-binding domain shared with APAF-1, various R-proteins, and CED-4; WT, wild type.

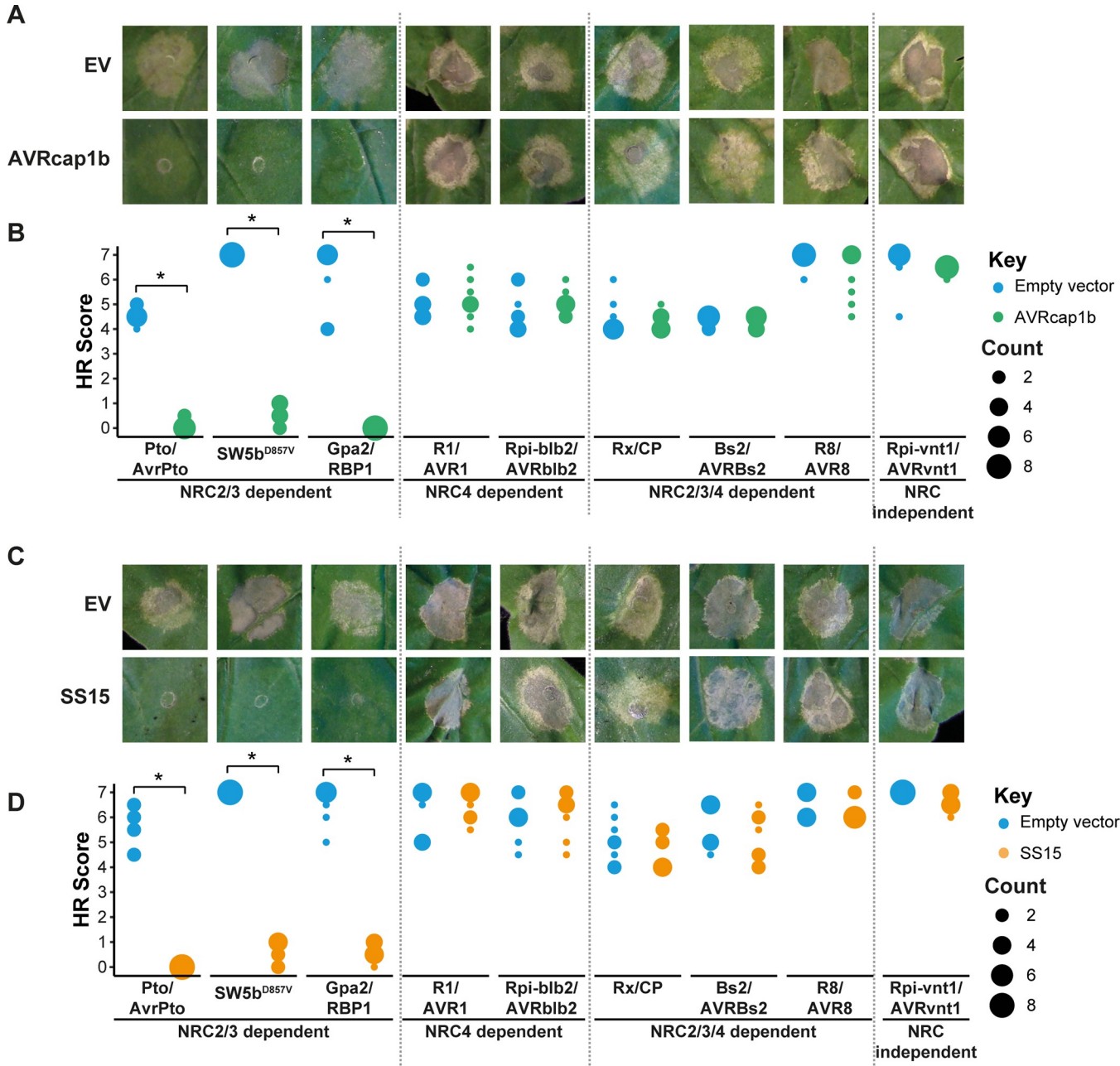

**Fig 3. AVRcap1b and SS15 suppress cell death mediated by sensor NLRs that signal through NRC2 and NRC3 but not NRC4.** Representative leaf panels showing HR phenotypes of (A) EV and AVRcap1b or (C) EV and SS15 coinfiltrated with a range of sensor NLRs (labelled on the bottom of panels B and D). Photographs were taken 5 days post-agroinfiltration. (B, D) HR results are presented as dot plots, where the size of each dot is proportional to the number of samples with the same score (count). A total of 8 technical replicates were completed for each treatment (S6 Data). Statistical tests were implemented using the besthr R library [41]. We performed bootstrap resampling tests using a lower significance cutoff of 0.025 and an upper cutoff of 0.975. Mean ranks of test samples falling outside of these cutoffs in the control samples bootstrap population were considered significant. Significant differences between the conditions are indicated with an asterisk (*). Details of statistical analysis are presented in S6 Fig. EV, empty vector; HR, hypersensitive response; NLR, nucleotide-binding domain and leucine-rich repeat; NRC, NLR required for cell death.

trailing necrotic lesions indicative of virus spread [24]. To determine the extent to which suppression mediated by AVRcap1b and SS15 translates into reduced viral infection, we tested the effectors ability to compromise Rx-mediated resistance to PVX. We transiently expressed

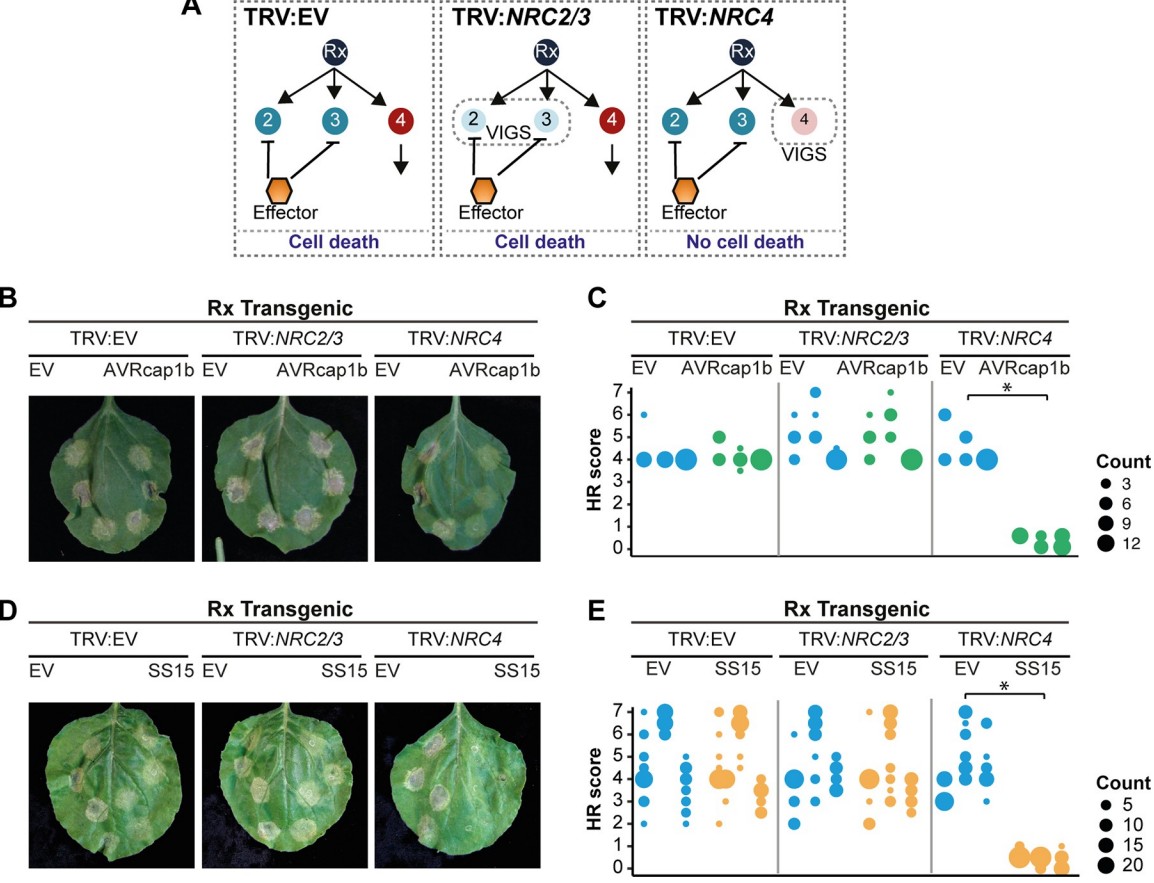

**Fig 4. AVRcap1b and SS15 suppress Rx-mediated cell death in *NRC4*-silenced Rx-transgenic *Nicotiana benthamiana* plants.** (A) A schematic representation of the silencing and infiltration strategy undertaken. Photo of representative *N. benthamiana* leaves showing HR after coexpression of (B) EV or AVRcap1b and (D) EV or SS15 in TRV:EV, TRV:*NRC2/3*, and TRV:*NRC4* silenced Rx-transgenic *N. benthamiana* plants, where EV was used as a negative control. Silencing treatments (TRV:EV, TRV:*NRC2/3*, and TRV:*NRC4*) are indicated above leaf panels. Plants were photographed 5 days after agroinfiltration. Cell death response was scored 5 days after agroinfiltration (S7 Data) (C) EV and AVRcap1b and (E) EV and SS15 and presented as dot plots, where dot colour represents either EV (blue), AVRcap1b (green), or SS15 (orange) and dot size is proportional to the number of samples with the same score (count). The experiment was independently repeated 3 times. Each replicate is represented by different columns within each silencing treatment for either EV, AVRcap1b, or SS15. Statistical tests were implemented using the besthr R library [41]. We performed bootstrap resampling tests using a lower significance cutoff of 0.025 and an upper cutoff of 0.975. Mean ranks of test samples falling outside of these cutoffs in the control samples bootstrap population were considered significant. Significant differences between the conditions are indicated with an asterisk (*). Details of statistical analysis are presented in S7 Fig. EV, empty vector; HR, hypersensitive response; VIGS, virus-induced gene silencing.

AVRcap1b or SS15 in Rx-transgenic *N. benthamiana* plants that were silenced for NRCs either individually (TRV:*NRC4*) or in combination (TRV:*NRC2/3*, TRV:*NRC2/3/4*). We then infected the leaves by agroinfection with *Agrobacterium tumefaciens* carrying PVX (pGR106:: PVX::GFP) using a toothpick inoculation method and documented the formation of trailing necrotic lesions at the inoculated spots (see PVX infection assays (agroinfection)) [24,43,46] (Fig 5A). Consistent with previous experiments [24], we observed trailing necrosis in the *NRC* triple silenced Rx leaves (TRV:*NRC2/3/4*) regardless of the presence or absence of AVRcap1b or SS15 (Fig 5B and 5C, S8 and S9 Data). We also observed trailing necrosis when AVRcap1b or SS15 was expressed in NRC4-silenced (TRV:*NRC4*) Rx leaves, indicating that both effectors compromised Rx-mediated resistance when NRC4 is depleted (Fig 5B and 5C, S8 and S9 Data). In contrast, Rx-mediated resistance to PVX was not compromised by AVRcap1b or

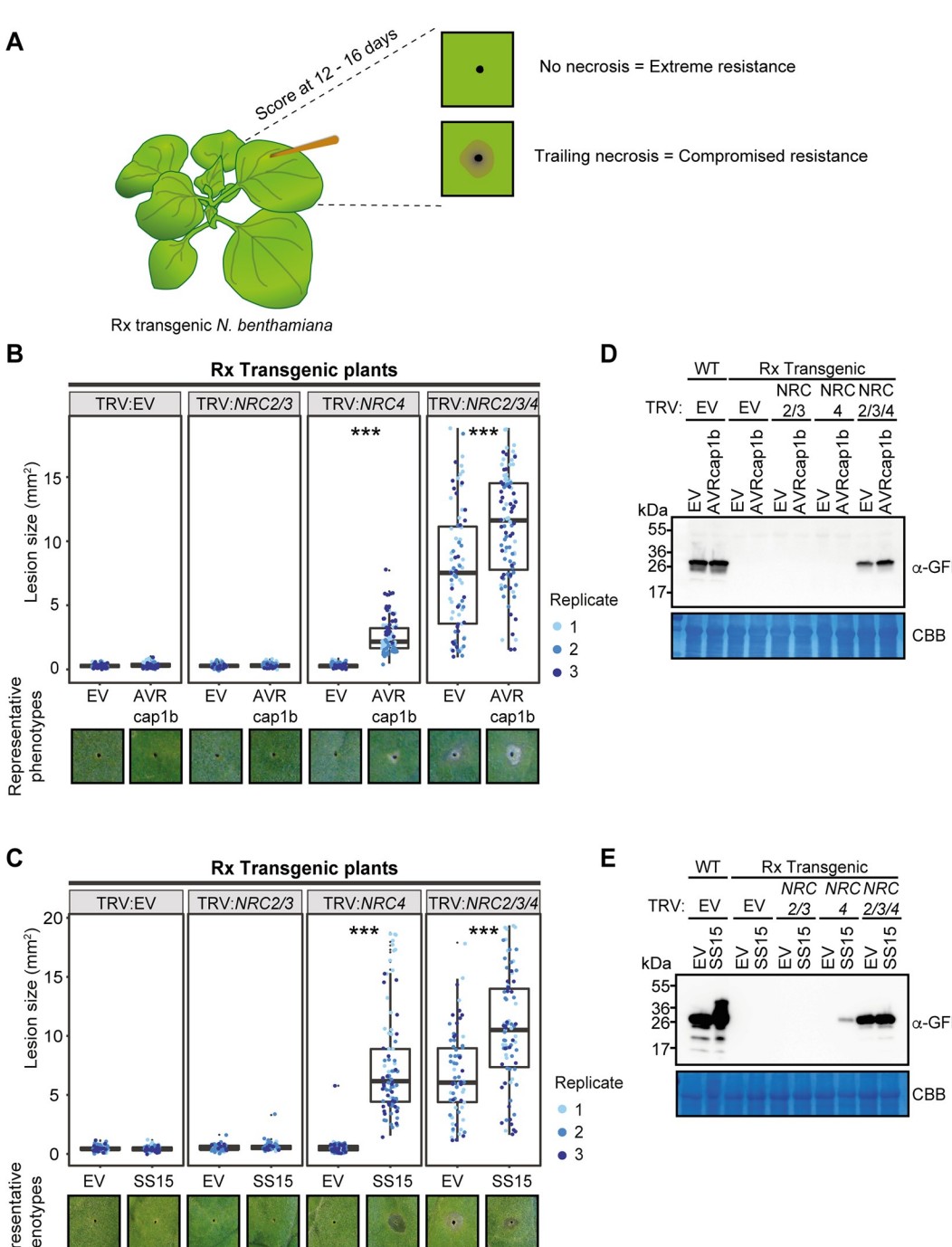

**Fig 5. AVRcap1b and SS15 compromise Rx-mediated extreme resistance to PVX in NRC4-silenced plants.** (A) Toothpick inoculation method previously described by Wu and colleagues allowed examination of the spread of trailing necrotic lesions due to the partial resistance mediated by Rx. EV, NRC2, NRC3, or NRC4 were silenced individually or in combination in Rx-transgenic *N. benthamiana* plants by TRV. (B) EV or AVRcap1b and (C) EV or SS15 were expressed, via agroinfiltration, in leaves of the NRC-silenced plants 1 day before PVX inoculation. PVX-GFP (pGR106-GFP) was inoculated using the toothpick inoculation method, as per panel (A). Photographs were taken 12–16 days after PVX inoculation. The size of the necrotic lesions was measured using Fiji (previously ImageJ) (S8 Data). Data acquired from different biological replications are presented in different colours. Statistical differences among the samples were analysed with mixed model ANOVA and a Tukey HSD test (*p*-value < 0.01), where the fixed effect is the silencing treatment (TRV: EV, TRV:*NRC2/3*, TRV:*NRC4*, TRV:*NRC2/3/4*) with either EV or AVRcap1b and the random effect is the experimental replicate. Significant differences between the conditions are indicated with asterisks (***, *p*-value < 0.0001) (S9 Data).

Representative phenotypes observed for each treatment are presented below the boxplots. Immunoblot analysis of GFP accumulation of pGR106-GFP toothpick inoculated sites in the presence of (D) AVRcap1b and (E) SS15 in Rx-transgenic or WT *N. benthamiana* plants. AVR, avirulence; CBB, Coomassie brilliant blue; EV, empty vector; HSD, honestly significant difference; NRC, NLR required for cell death; PVX, Potato virus X; TRV, Tobacco rattle virus; WT, wild type.

SS15 in *NRC2/3*-silenced (TRV:*NRC2/3*) leaves. Virus accumulation in the different treatments was validated by western blot detection of GFP protein driven by the subgenomic promoter of PVX::GFP (Fig 5D and 5E). Perturbation of PVX resistance was more markedly affected by SS15 compared to AVRcap1b based on lesion size and PVX accumulation (Fig 5C–5E). These results indicate that the AVRcap1b and SS15 effectors not only suppress NLR-mediated cell death but also counteract the disease resistance phenotype mediated through NRC2 and NRC3.

## Yeast two-hybrid screens reveal candidate host interactors of the AVRcap1b and SS15 effectors

To investigate how AVRcap1b and SS15 suppress NRC activities, we set out to identify their host interactors. We used the effectors as baits in unbiased yeast two-hybrid (Y2H) screens against a *N. benthamiana*–mixed tissue cDNA library (ULTImate Y2H, Hybrigenics Services, Paris, France). AVRcap1b was screened against a combined approximately 140 million clones, which resulted in 13 candidate interacting proteins from a total of 35 positive clones (S2 Table). SS15 was screened against approximately 61 million clones resulting in 10 candidate proteins from a total of 202 positive clones (S3 Table). Remarkably, NRC3 and NRC4 were among the recovered SS15 protein interactors, which is notable given that NLR proteins are rarely recovered from Y2H screens. The NRC3 and NRC4 fragments that were recovered from the Y2H screen matched the CC-NB-ARC domains indicating that SS15 may bind the N-terminal half of NRC proteins (Fig 6A, S3 Table). NRC2 was not recovered as an interactor in the Y2H screen for SS15; however, we cannot rule out that this might be due to poor accumulation of this NLR or its subdomains in the yeast strains used.

To further investigate the interactions between SS15 and NRCs, we sought out to delimit the binding domain within the NLRs. Since the prey fragment hits from the Y2H screen covered the CC-NB-ARC domain of NRC3 and NRC4 (Fig 6A), we generated CC and NB-ARC domain truncations and tested them in additional Y2H experiments for interaction with SS15. These assays revealed that SS15 binds to the NB-ARC but not the CC domains of NRC2, NRC3, and NRC4 (S8 Fig). Consistent with earlier observations, AVRcap1b did not interact with either the CC or NB-ARC domains of the NRCs in these Y2H experiments (S8 Fig).

## Unlike AVRcap1b, SS15 associates with NRC2 and NRC3 *in planta*

The Y2H results prompted us to investigate the association between AVRcap1b, SS15, and NRCs using co-immunoprecipitation (coIP) of proteins expressed in *N. benthamiana*, which should reveal the association between the effectors and full-length NRC proteins in a more physiologically relevant condition. To achieve this, we coexpressed each of AVRcap1b::6xHA and 4xHA::SS15 with NRC2::4xMyc, NRC3::4xMyc, and NRC4::4xMyc in *N. benthamiana* leaves using agroinfiltration and subjected the protein extracts to anti-Myc immunoprecipitation and western blot analyses. These experiments revealed that SS15 co-immunoprecipitated with NRC2 and NRC3, but not with NRC4 (Fig 6B). While we did detect association between NRC4 and SS15 in Y2H, we could only detect a weak NRC4 signal in some biological replicates, suggesting that SS15 may also associate with NRC4 *in planta* but with markedly lower

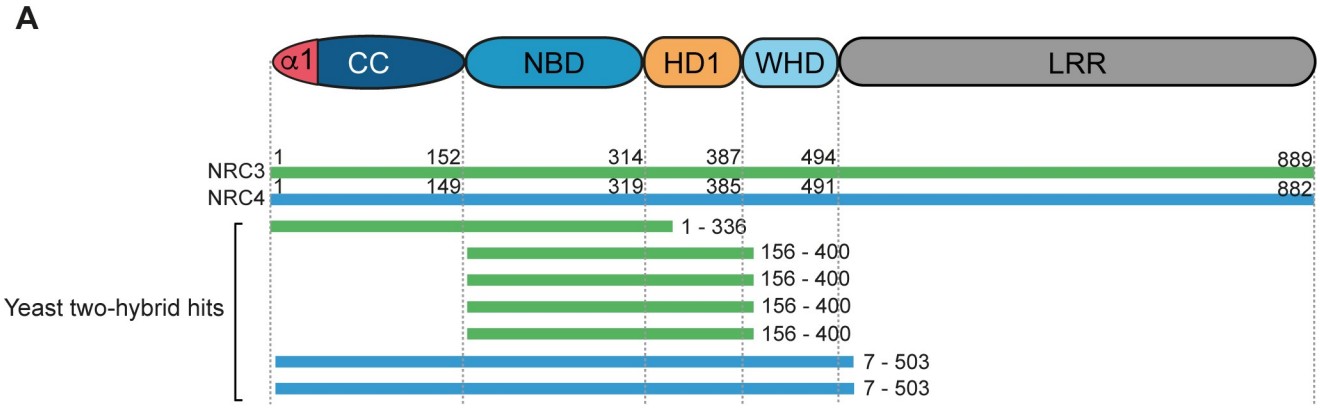

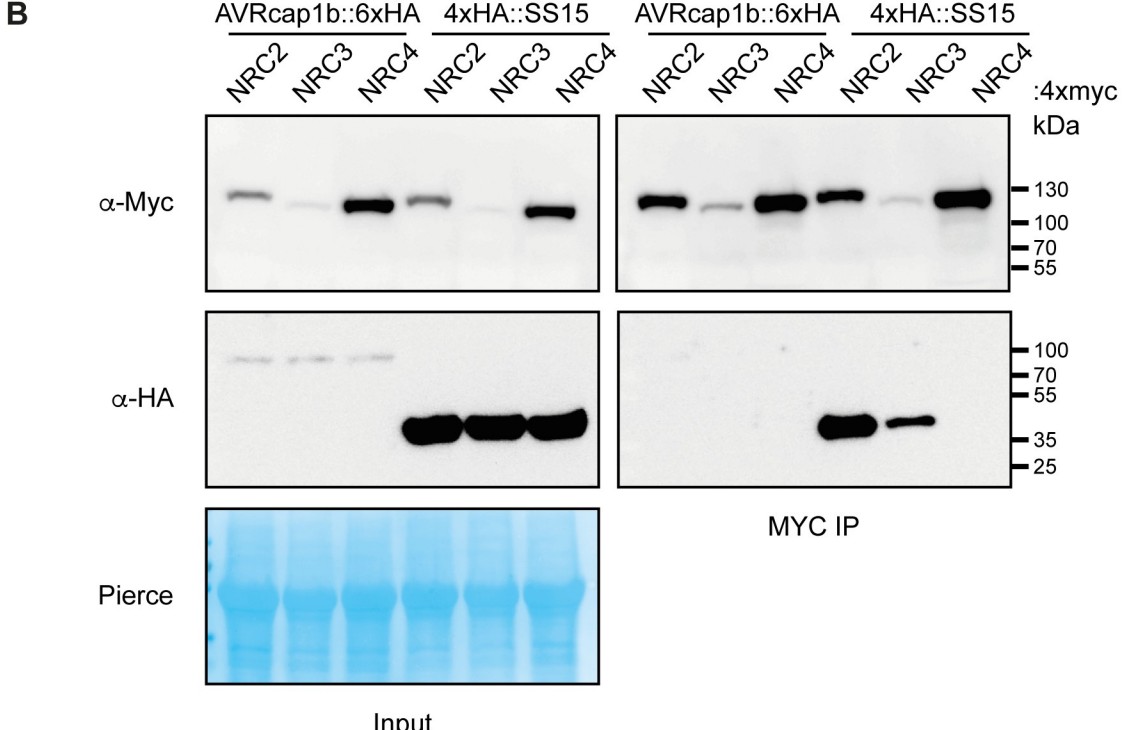

**Fig 6. SS15, but not AVRcap1b, associates with NRCs *in planta*.** (A) Schematic diagram of domain organisation of NRC proteins, showing Y2H hits from the Hybrigenics services screen (S3 Table). (B) CoIP experiment of C-terminally 4xmyc-tagged NRC2, NRC3, and NRC4 with C-terminally HA-tagged AVRcap1b::6xHA and N-terminally tagged 4xHA:SS15 (labelled above). Proteins obtained by coIP with MYC beads (MYC IP) and total protein extracts (input) were immunoblotted with the appropriate antisera labelled on the left. Approximate molecular weights (kDa) of the proteins are shown on the right. Rubisco loading control was carried out using Pierce staining. The experiment was performed more than 3 times under different pulldown conditions with similar results. CC, coiled-coil; coIP, co-immunoprecipitation; HD1, helical domain 1; LRR, leucine-rich repeat; NBD, nucleotide-binding domain; NRC, NLR required for cell death; WHD, winged helix domain; Y2H, yeast two-hybrid.

affinity compared to NRC2 and NRC3 (S9 Fig). This result aligns with our observation that SS15 is unable to suppress NRC4 in cell death assays. In the case of AVRcap1b, the coIP experiments were consistent with the Y2H screens as we did not detect an association between AVRcap1b and any of the 3 NRCs (Fig 6B).

## SS15 directly binds the NB-ARC domain of NRC3 in vitro

To further examine the association between SS15 and the NB-ARC domains of NRCs, we purified these proteins for in vitro assays using *Escherichia coli* as a heterologous expression system. We successfully obtained homogeneous SS15 and NRC3 NB-ARC domain (NRC3^NB-ARC) proteins but were unable to obtain purified NRC2 NB-ARC or NRC4 NB-ARC domains due to solubility and stability issues. We subjected purified SS15 and NRC3^NB-ARC to gel filtration and found that these proteins elute at 239 ml and 227 ml, which correspond to 42.52 and 27.75 kDa, respectively (Fig 7). To determine whether the 2 proteins form a complex in vitro, we mixed cells expressing individual proteins and copurified SS15 and NRC3^NB-ARC for gel filtration assays (see Protein–protein interaction studies: Protein purification from *E. coli* and in vitro protein–protein interaction studies). The copurified mixture of SS15 and NRC3^NB-ARC resulted in a peak shift with an elution volume at 211 ml, which corresponds to 84.02 kDa. Further validation by SDS-PAGE of the fractions under the new peak confirmed the presence of both proteins (Fig 7). In our gel filtration assays, the protein molecular weight calibration led to overestimates of the predicted molecular masses of the proteins, both alone and in complex (NRC3^NB-ARC is 41 kDa, SS15 is 24.5 kDa, and NRC3^NB-ARC–SS15 complex is 65.5 kDa). However, the results indicate that monomeric forms of each state exist in solution and that the 2 proteins probably enter in a 1:1 complex under these experimental conditions. Taken together, these results suggest that SS15 forms a complex with NRC^NB-ARC in vitro.

## SS15 associates with p-loop mutants of NRC2 and NRC3

The p-loop motif within the NB-ARC domain of NLR proteins is crucial for ATP binding and hydrolysis, a biochemical step that is essential for NLR oligomerisation and activation [47]. To determine whether SS15 associates with p-loop mutants of the NRCs, we performed coIP experiments in *N. benthamiana* with the NRC p-loop mutants NRC2^K188R, NRC3^K191R, and NRC4^K190R [24]. These experiments revealed that SS15 associates with the p-loop mutants of NRC2 (NRC2^K188R) and NRC3 (NRC3^K191R) (S10 Fig). Since an intact p-loop is not required for SS15-NRC association, our findings suggest that SS15 can probably enter in complex with inactive forms of NRC2 and NRC3.

## SS15 associates with activated forms of NRC2

We next investigated the extent to which SS15 associates with activated NRC2 and NRC3. We coexpressed 4xHA::SS15 with C-terminally 4xMyc-tagged autoimmune forms of NRC2^H480R and NRC3^D480V in *N. benthamiana* using agroinfiltration and subjected the protein extracts to anti-Myc immunoprecipitation and western blot analyses. We found that SS15 co-immunoprecipitated with NRC2^H480R, indicating that this effector associates with activated forms of NRC2 (Fig 8). Our results for NRC3^D480V, however, were inconclusive, since this protein displayed poor accumulation and could not be detected in western blot analyses under these conditions. Despite this, our overall conclusion is that SS15 binds both inactive and activated forms of NRC2.

## AVRcap1b associates with the *Nicotiana benthamiana* ENTH/VHS-GAT domain protein TOL9a *in planta*

Given that AVRcap1b did not interact with NRCs, we reasoned that this effector targets another host protein that is involved in NLR immunity. To narrow down the list of 13 candidate interactors obtained from the Y2H screen, we subjected AVRcap1b to *in planta* coIP coupled with tandem mass spectrometry (IP-MS) using methods that are well established in our

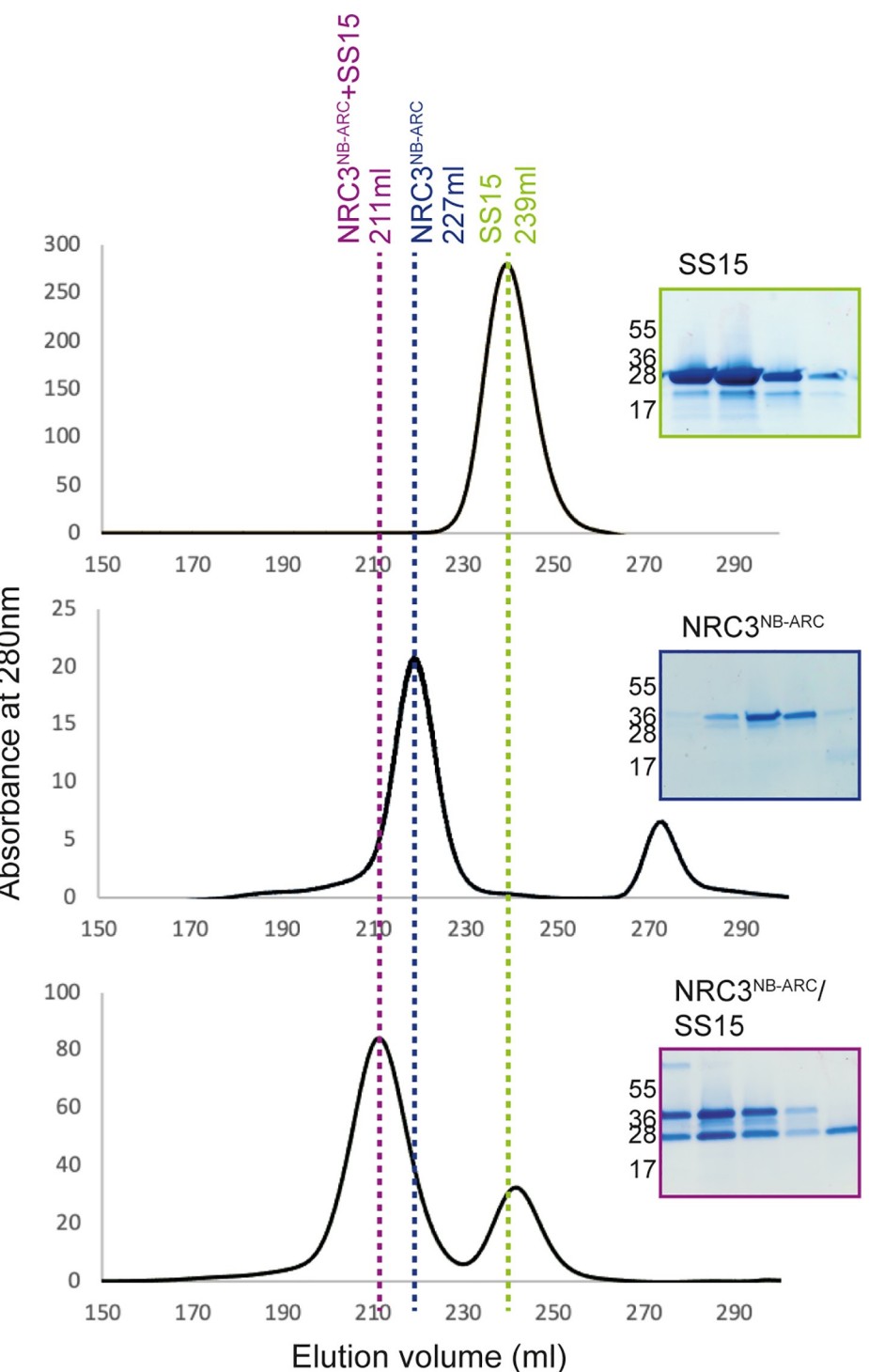

**Fig 7. SS15 binds the NB-ARC domain of NRC proteins in vitro.** SS15 binds the NRC3[NB-ARC] domain in vitro. Gel filtration traces obtained for SS15 (top), NRC3[NB-ARC] domain (middle), and a 1:1 mixture of the complex (bottom). Insets show SDS-PAGE gels of the fractions collected across the elution peaks.

laboratory [48–50]. IP-MS experiments with GFP::AVRcap1b resulted in the identification of 8 unique AVRcap1b interactors that were not recovered with the control GFP::PexRD54, another *P. infestans* effector of a similar fold and size (S11 Fig, S4 Table). Three of the 8

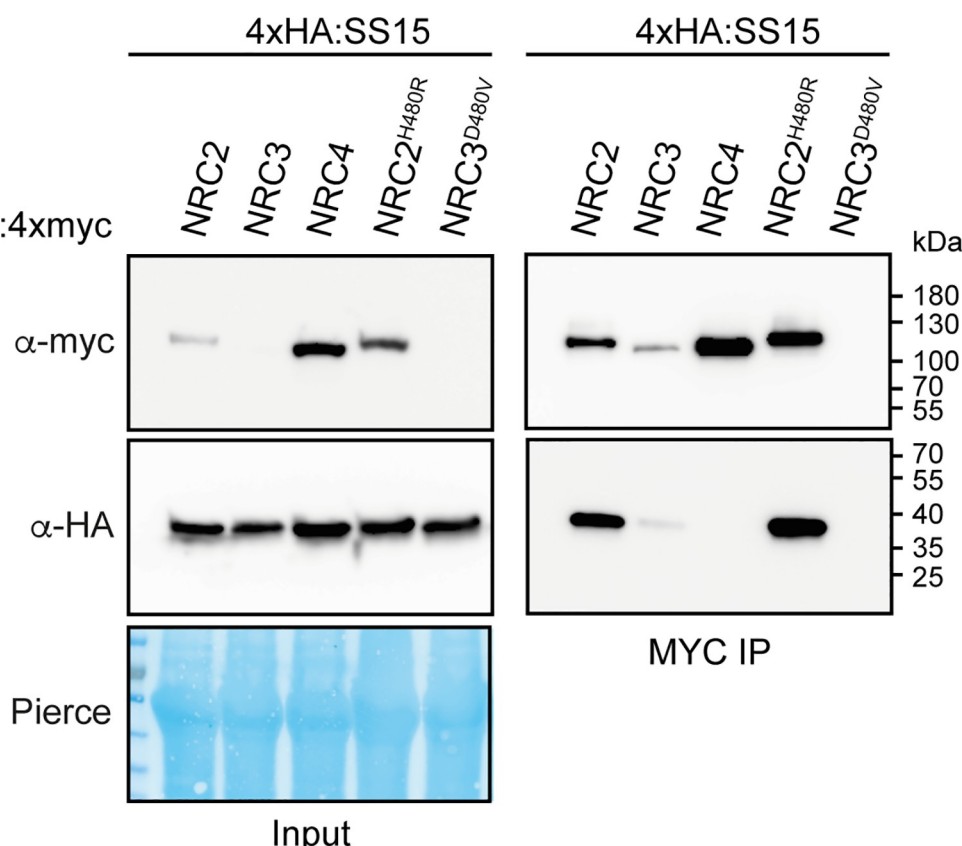

**Fig 8. SS15 associates with autoactive NRC2.** N-terminally 4xHA-tagged SS15 was transiently coexpressed in *N. benthamiana* with C-terminally 4xMyc tagged-NRC2, NRC3, NRC4, NRC2$^{H480R}$, NRC3$^{D480V}$, and NRC4$^{D478V}$. IP was performed with agarose beads conjugated to Myc (Myc IP) antibodies. Total protein extracts (Input) and proteins obtained by coIP were immunoblotted with appropriate antisera labelled on the right. Approximate molecular weights (kDa) of the proteins are shown on the right. Rubisco loading controls were conducted using Pierce staining. This experiment is representative of 2 independent replicates. coIP, co-immunoprecipitation; IP, immunoprecipitation.

interactors were different members of the Target of Myb 1-like protein (TOL) family of ENTH/VHS-GAT domain-containing proteins that function in membrane trafficking as part of the endosomal sorting complex required for transport (ESCRT) pathway [51,52] (Figs 9A and S10 and S4 Table). One of these candidate TOLs, Nbv6.1trP4361, was independently recovered in the Y2H screen (S2 Table). Therefore, we decided to further investigate the corresponding protein (hereafter referred to as NbTOL9a) as a candidate host target of AVRcap1b.

Computational analyses of the *N. benthamiana* genome revealed 4 TOL paralogs in addition to NbTOL9a, which we termed NbTOL9b (Nbv6.1trP9166), NbTOL3 (Nbv6.1trA40123), NbTOL6 (Nbv6.1trP73492), and NbTOL9c (Nbv6.1trA64113) following previously published nomenclature (S5 Table, S12 Fig) [53]. To validate the association between AVRcap1b and NbTOL proteins, we coexpressed GFP::AVRcap1b with C-terminally 6xHA-tagged fusions of the 5 TOL paralogs, in *N. benthamiana* leaves, and performed anti-GFP and anti-HA immunoprecipitations. AVRcap1b associated with NbTOL9a and, to a lesser extent, with NbTOL9b and NbTOL9c in the GFP pulldown (GFP IP). However, AVRcap1b only associated with NbTOL9a in the reciprocal HA pulldown (HA IP) (Fig 9B). In both experiments, NbTOL9a protein did not associate with the negative control GFP::PexRD54. These results indicate that AVRcap1b associates with members of the NbTOL family, exhibiting a stronger affinity for NbTOL9a. Based on this conclusion, we focused subsequent experiments on NbTOL9a.

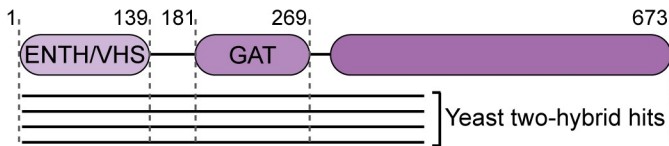

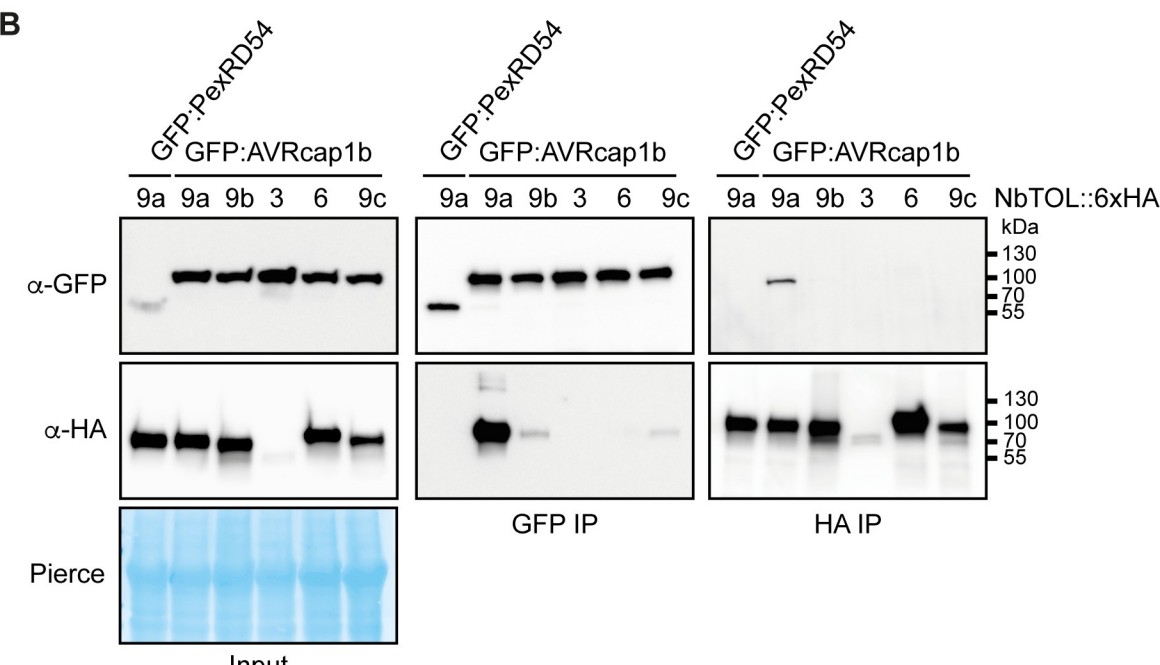

**Fig 9. AVRcap1b associates with NbTOL9a *in planta*.** (A) Schematic diagram of domain organisation of NbTOL9a, showing Y2H hits from the screen (S2 Table). (B) CoIP experiment between AVRcap1b and 5 NbTOL family proteins (NbTOL9a, NbTOL9b, NbTOL3, NbTOL6, and NbTOL9c). N-terminally GFP-tagged AVRcap1b was transiently coexpressed with all 5 NbTOL proteins fused to a C-terminal 6xHA tag. N-terminally GFP-tagged PexRD54 was used as a negative control. IP were performed with agarose beads conjugated to either GFP (GFP-IP) or HA (HA-IP) antibodies. Total protein extracts were immunoblotted with appropriate antisera labelled on the left. Approximate molecular weights (kDa) of the proteins are shown on the right. Rubisco loading controls were conducted using Pierce staining. This experiment is representative of 3 independent replicates. coIP, co-immunoprecipitation; IP, immunoprecipitation; NbTOL, *N. benthamina* Target of Myb 1-like protein; Y2H, yeast two-hybrid.

## NbTOL9a negatively modulates the cell death triggered by NRC2 and NRC3 but not NRC4 or NbZAR1

To gain additional insights into the role of NbTOL9a in NRC-mediated hypersensitive cell death, we altered NbTOL9a expression in *N. benthamiana*. First, we investigated the effect of silencing *NbTOL9a* on NRC autoimmunity. We generated a hairpin-silencing construct (RNAi::*NbTOL9a*) that mediates silencing of *NbTOL9a* in transient expression assays in *N. benthamiana* leaves (S13 Fig). We then coexpressed RNAi::*NbTOL9a* with NRC2$^{H480R}$ and NRC3$^{D480V}$ using agroinfiltration of *N. benthamiana* leaves to test the degree to which silencing of *NbTOL9a* affects NRC2- and NRC3-mediated cell death. To improve robustness of the assay, we used increasing concentrations of *A. tumefaciens* expressing NRC2$^{H480R}$ and NRC3$^{D480V}$ (OD$_{600}$ = 0.1, 0.25, or 0.5). Silencing of *NbTOL9a* at all tested OD$_{600}$ concentrations enhanced the cell death response triggered by NRC2$^{H480R}$ and NRC3$^{D480V}$ but did not

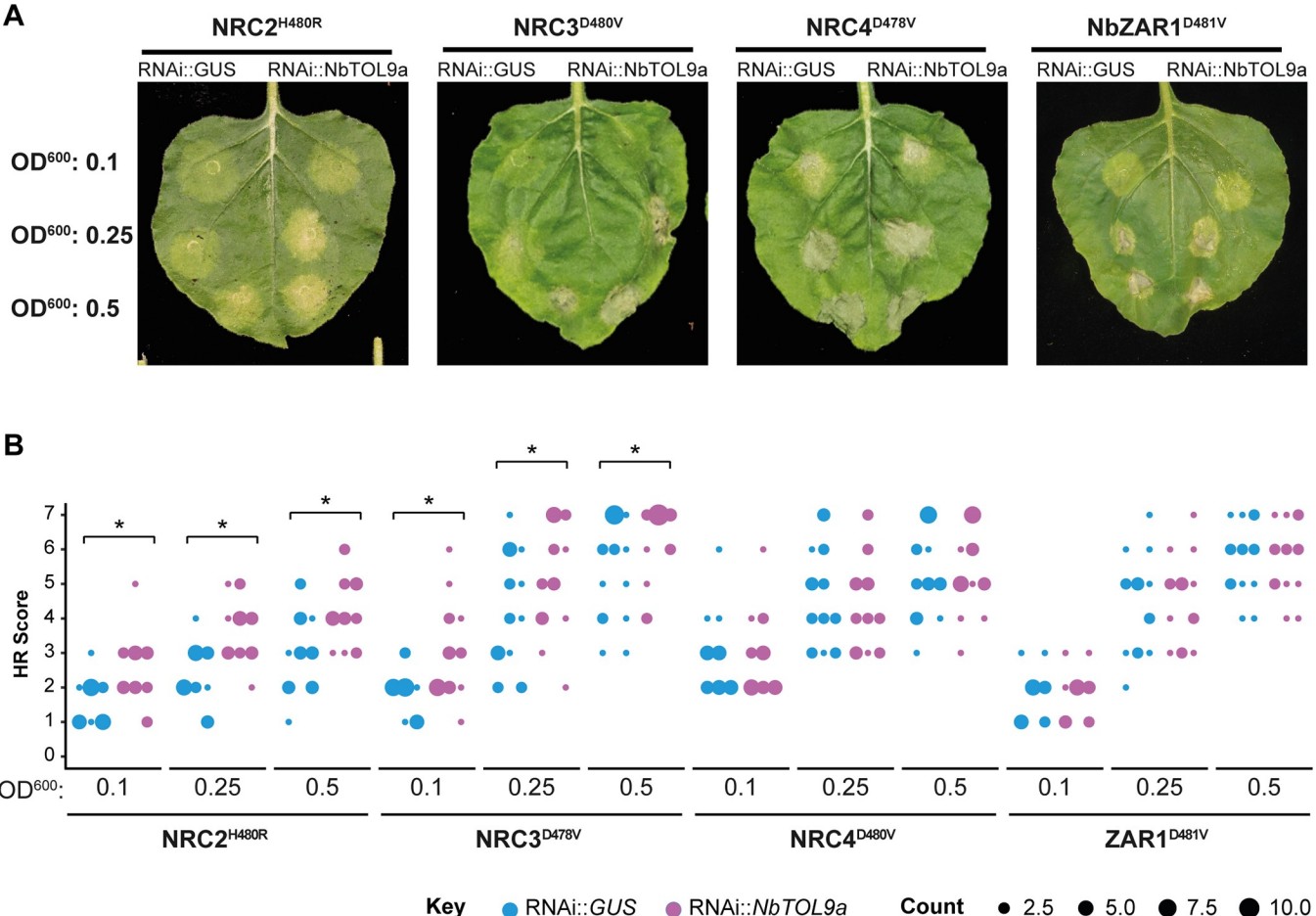

**Fig 10. Silencing of NbTOL9a enhances cell death mediated by NRC2$^{H480R}$ and NRC3$^{D480V}$ but not NbZAR1$^{D481V}$ or NRC4$^{D478V}$.** (A) Photo of representative *N. benthamiana* leaves showing HR after coexpression of NRC2$^{H480R}$, NRC3$^{D480V}$, NRC4$^{D478V}$, and NbZAR1$^{D481V}$ with RNAi::*GUS* (control) and RNAi:*NbTOL9a* (labelled above leaf panels). To improve the robustness of the assay, we used increasing concentrations of *A. tumefaciens* expressing NRC2$^{H480R}$, NRC3$^{D480V}$, NRC4$^{D478V}$, and NbZAR1$^{D481V}$ (OD$_{600}$ = 0.1, 0.25, or 0.5) (S10 Data). HR response was scored and photographed 5 days after agroinfiltration. (B) HR results are presented as dot plots, where the size of each dot is proportional to the number of samples with the same score (count). Three biological replicates were completed, indicated by columns for RNAi::*GUS* and RNAi::*NbTOL9a*, for each treatment combination. Statistical tests were implemented using the besthr R library [41]. We performed bootstrap resampling tests using a lower significance cutoff of 0.025 and an upper cutoff of 0.975. Mean ranks of test samples falling outside of these cutoffs in the control samples bootstrap population were considered significant. Significant differences between the conditions are indicated with an asterisk (*). The details of statistical analysis are presented in (S14 Fig). HR, hypersensitive response.

affect NRC4$^{D478V}$ or NbZAR1$^{D481V}$ (NRC-independent NLR; [54]), compared to the RNAi::*GUS* silencing control (Figs 10 and S14 and S10 Data).

Next, we determined the effect of NbTOL9a overexpression by coexpressing it with the autoimmune NRCs in *N. benthamiana* leaves using agroinfiltration. Since NRC2$^{H480R}$ autoimmune response is comparably weaker than NRC3$^{D480V}$, we focused on NRC3$^{D480V}$ in this and subsequent experiments as it provides a more robust readout for cell death–based assays. NbTOL9a overexpression reduced the cell death response triggered by NRC3$^{D480V}$ but did not affect NRC4$^{D478V}$ or the constitutively active MEK2$^{DD}$, a mitogen-activated protein kinase kinase (MAPKK) involved in plant immune signalling, which we included as an additional control (Figs 11 and S15 and S11 Data). Altogether, these 2 sets of experiments indicate that NbTOL9a modulates NRC2 and NRC3 activities in a manner consistent with a negative regulatory role in NRC2- and NRC3-mediated immunity.

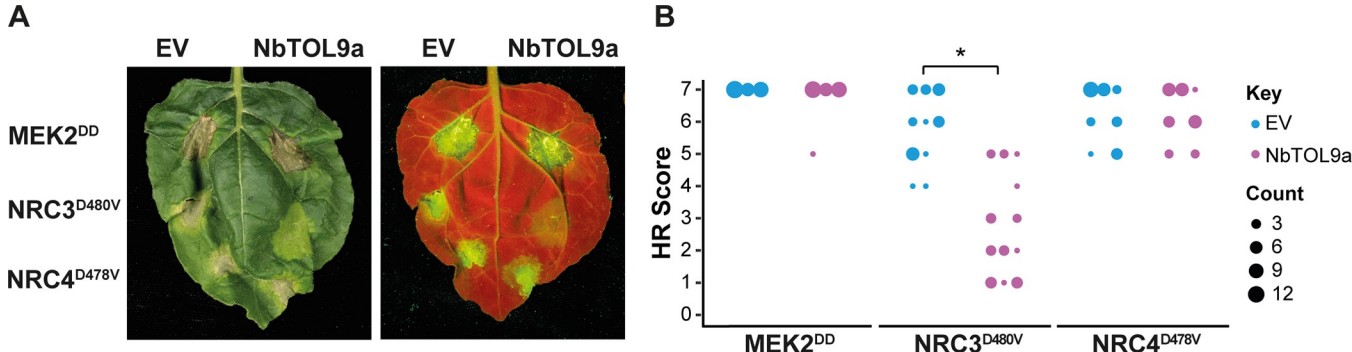

**Fig 11. Overexpression of NbTOL9a suppresses autoactive NRC3$^{D480V}$ but not MEK2$^{DD}$ or NRC4$^{D478V}$.** (A) Photo of representative *N. benthamiana* leaves showing HR after coexpression of EV and NbTOL9a (labelled above leaf panels) with MEK2$^{DD}$, NRC3$^{D480V}$, and NRC4$^{D478V}$. HR response was scored and photographed 5 days after agroinfiltration (left panel under white light, right panel autofluorescence under UV light). MEK2$^{DD}$ was included as a positive control for cell death (S11 Data). (B) HR results are presented as dot plots, where the size of each dot is proportional to the number of samples with the same score (count). Three biological replicates were completed, indicated by columns for EV, NbTOL9a in each treatment (MEK2$^{DD}$, NRC3$^{D480V}$, NRC4$^{D478V}$). Statistical tests were implemented using the besthr R library [41]. We performed bootstrap resampling tests using a lower significance cutoff of 0.025 and an upper cutoff of 0.975. Mean ranks of test samples falling outside of these cutoffs in the control samples bootstrap population were considered significant. Significant differences between the conditions are indicated with an asterisk (*). Details of statistical analysis are presented in (S15 Fig). EV, empty vector; HR, hypersensitive response.

## AVRcap1b suppression of NRC3 is compromised in the absence of NbTOL9a

Our finding that NbTOL9a exhibits a negative regulatory role in NRC2- and NRC3-mediated cell death led us to the hypothesis that AVRcap1b is co-opting NbTOL9a to execute its suppression activity. To test this, we coexpressed AVRcap1b with the autoimmune mutants NRC3$^{D480V}$ or NRC4$^{D478V}$ in *N. benthamiana* leaves that are expressing either RNAi:: *NbTOL9a* (*NbTOL9a* silenced) or RNAi::*GUS* (negative control). Consistent with Fig 2, overexpression of AVRcap1b suppressed the cell death triggered by NRC3$^{D480V}$ but not by NRC4$^{D478V}$ (Fig 12, S12 Data). However, silencing of *NbTOL9a* compromised AVRcap1b suppression of NRC3$^{D480V}$ autoimmunity and partially restored the cell death phenotype (Fig 12, S12 Data). These results indicate that AVRcap1b genetically requires NbTOL9a to fully downregulate NRC3 cell death activity and is likely co-opting this host protein to execute its suppression activities.

## NbTOL9a does not associate with NRCs *in planta*

Given that NbTOL9a negatively regulates NRC2 and NRC3, we hypothesised that NbTOL9a associates with NRC2 and NRC3 to execute its immunomodulatory functions. Furthermore, since AVRcap1b requires NbTOL9a to fully suppress NRC3, AVRcap1b may act as a suppressor through altering NbTOL9a–NRC associations. To test these hypotheses, we coexpressed NbTOL9a::6xHA with NRC2::4xMyc, NRC3::4xMyc, and NRC4::4xMyc, either in the presence of GFP::AVRcap1b or free GFP in *N. benthamiana* leaves using agroinfiltration, and performed anti-Myc immunoprecipitation followed by western blot analyses. We included AVRcap1b::4xMyc as a positive control for association with NbTOL9a. While we were able to successfully detect association between NbTOL9a:6xHA and AVRcap1b::4xMyc, we did not observe any association between NbTOL9a:6xHA and any of the 3 NRC proteins tested, regardless of whether GFP::AVRcap1b was present or absent (Fig 13). These results suggest that NbTOL9a-mediated negative regulation of NRC2 and NRC3 does not involve association between these 2 proteins *in planta* or, alternatively, involves protein–protein interactions that

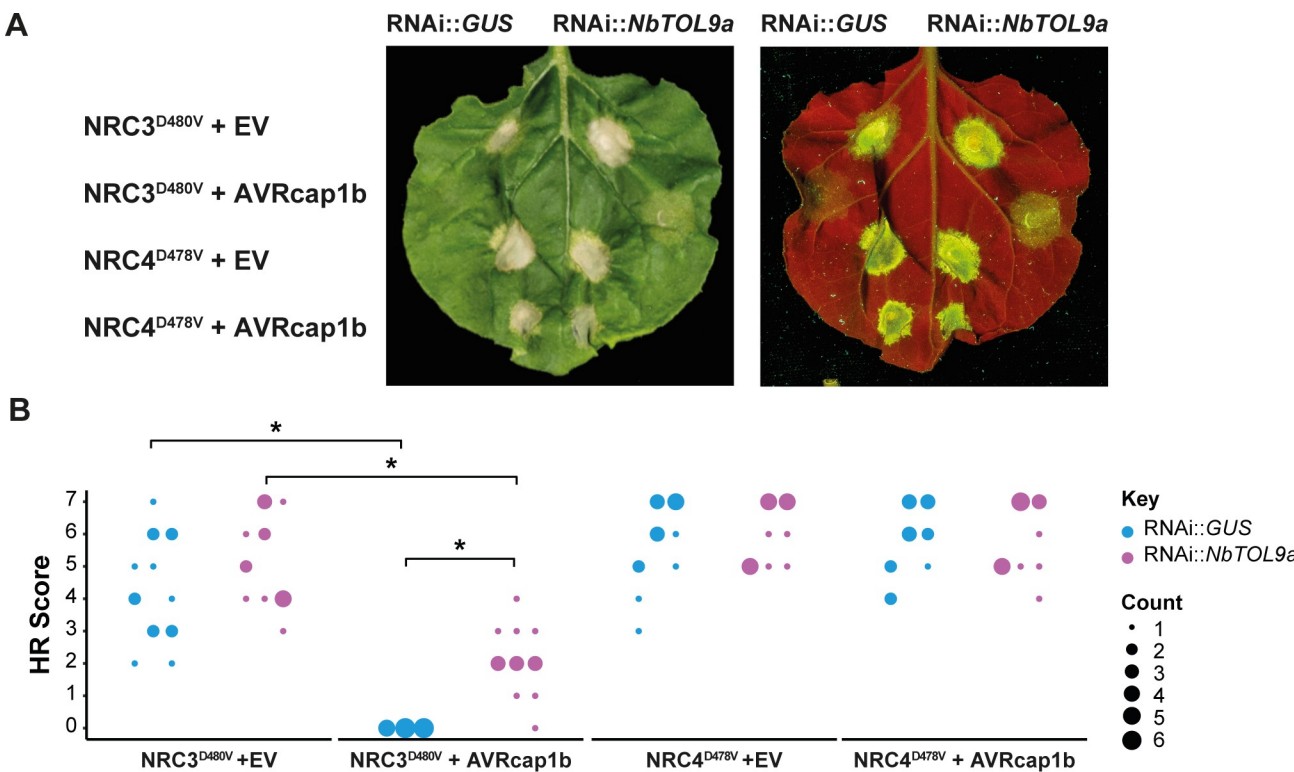

**Fig 12. Silencing of NbTOL9a compromises AVRcap1b-mediated suppression of NRC3.** (A) Photo of representative *N. benthamiana* leaves showing HR after coexpression of RNAi::*GUS* and RNAi::*NbTOL9a* with NRC3$^{D480V}$ + EV, NRC3$^{D480V}$ + AVRcap1b, NRC4$^{D478V}$ + EV, and NRC4$^{D478V}$ + AVRcap1b. HR response was scored and photographed 5 days after agroinfiltration (left panel under white light, right panel autofluorescence under UV light) (S12 Data). (B) HR results are presented as dot plots, where the size of each dot is proportional to the number of samples with the same score (count). Results are based on 3 biological replicates. Statistical tests were implemented using the besthr R library [41]. We performed bootstrap resampling tests using a lower significance cutoff of 0.025 and an upper cutoff of 0.975. Mean ranks of test samples falling outside of these cutoffs in the control samples bootstrap population were considered significant. Significant differences between the conditions are indicated with an asterisk (*). Details of statistical analysis are presented in (S16 Fig). HR, hypersensitive response.

are too transient to be detected by coIP. Moreover, we conclude that AVRcap1b does not alter NbTOL9a–NRC associations (or lack thereof) to execute its immune suppression activities.

## Discussion

The aim of this study was to address the hypothesis that solanaceous parasites have evolved effector proteins that target the NRC network of NLR immune receptors. We confirmed this hypothesis by carrying out an effectoromics screen, which yielded 5 effectors that can compromise the NRC network: SS10, SS15, and SS34 from the cyst nematode *G. rostochiensis* and AVRcap1b and PITG-15278 from the potato late blight pathogen *P. infestans*. These 5 effectors can suppress the hypersensitive cell death induced in *N. benthamiana* by either Prf or Rpi-blb2, 2 NRC-dependent sensor NLRs that function as bona fide disease resistance proteins. Interestingly, these effectors appear to function at different points in the NRC network (Fig 14). While SS10, SS34, and PITG-15278 suppress cell death mediated by Rpi-blb2, they do not interfere with an autoimmune mutant of the downstream helper NRC4. SS15 and AVRcap1b, however, can robustly suppress autoimmune mutants of NRC2 and NRC3, indicating that they act at the level of the NRC helpers or their downstream pathways. We found that SS15 directly binds the NB-ARC domain of NRC2 and NRC3, while AVRcap1b associates with NbTOL9a and requires this host protein to fully suppress NRC3. We conclude that cyst

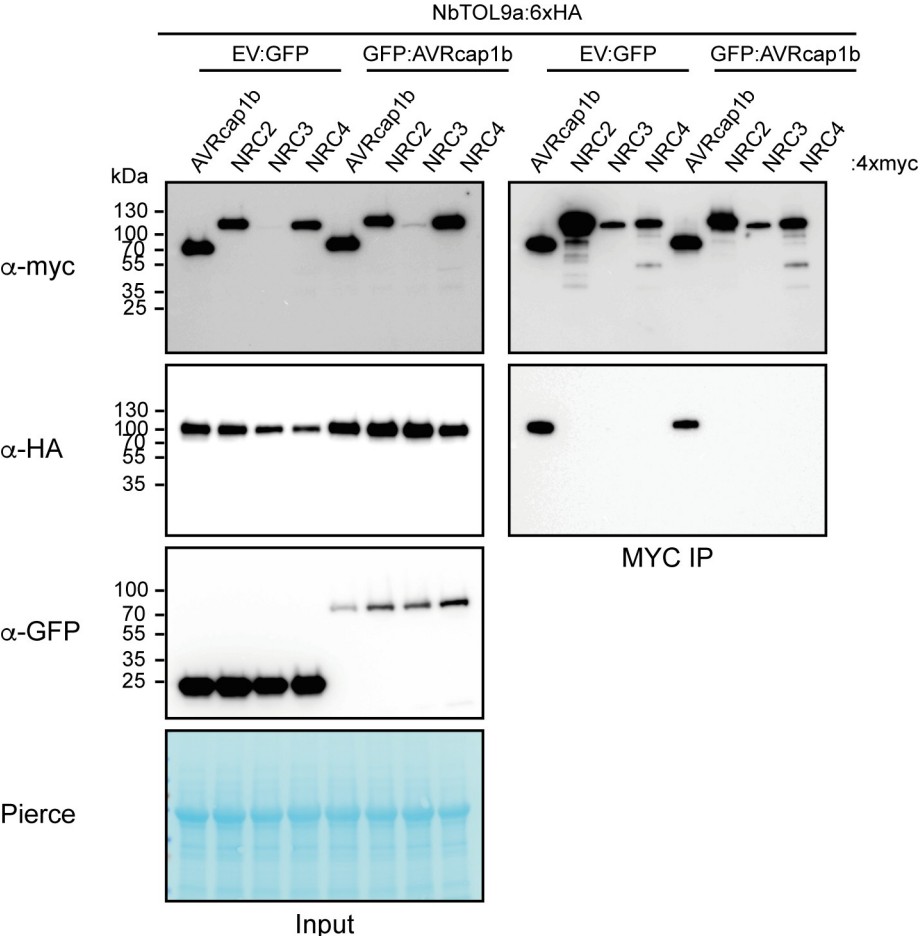

**Fig 13. NbTOL9a does not associate with NRCs *in planta*.** CoIP experiment between NbTOL9a and NRCs in the presence and absence of AVRcap1b. C-terminally 6xHA-tagged NbTOL9a was transiently coexpressed with NRC2, NRC3, and NRC4 proteins fused to a C-terminal 4xmyc tag in the presence of free GFP (EV:GFP) or N-terminally GFP-tagged AVRcap1b. C-terminally 4xmyc-tagged AVRcap1b was used as a positive control for association with NbTOL9a. IP was performed with agarose beads conjugated to MYC (MYC-IP) antibodies. Total protein extracts were immunoblotted with appropriate antisera labelled on the left. Approximate molecular weights (kDa) of the proteins are shown on the left. Rubisco loading controls were conducted using Pierce staining. This experiment is representative of 2 independent replicates. coIP, co-immunoprecipitation; IP, immunoprecipitation; NRC, NLR required for cell death.

nematodes and *P. infestans* convergently evolved sequence unrelated effectors that target key nodes of the NRC network or their downstream components to suppress host immune signalling. Our paper also highlights the value of using effectors as probes to dissect key regulatory components of immunity and to study the complex interactions between NLR receptors and the networks they form.

Our findings help to explain why plants have evolved NLR receptor networks with complex architectures. We previously postulated that NLR networks, such as the NRC network, help maintain the robustness of the immune system in light of external perturbations [55]. A key feature of the NRC network is that NRCs act as central nodes that function downstream of multiple disease resistance proteins (NLR sensors) that form a massively expanded phylogenetic clade in the Solanaceae. The NRC nodes have overlapping NLR sensor specificities and display varying degrees of redundancy [24]. Given that NRCs are critical for immune signalling of multiple disease resistance proteins, they would be ideal targets for pathogen effectors.

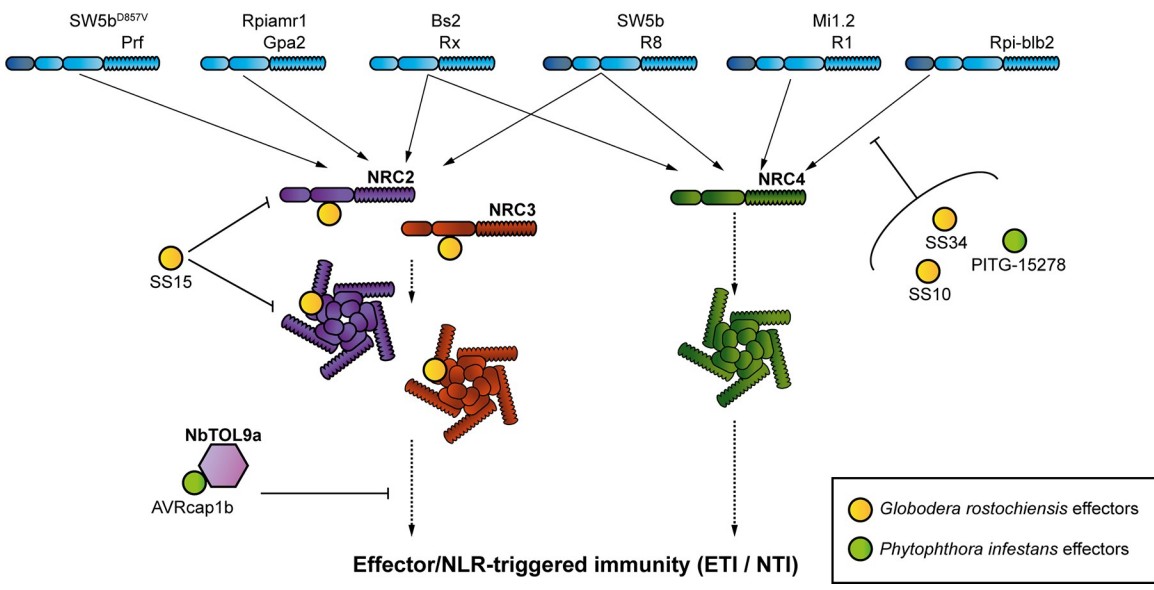

**Fig 14. Evolutionary divergent pathogens have evolved to target multiple layers of the Solanaceae NLR network.** *P. infestans* and *G. rostochiensis*, 2 evolutionary distinct pathogens, have evolved effectors that suppress signalling mediated by the NRC network. Two effectors from the cyst nematode pathogen, *G. rostochiensis* (SS10 and SS34), and 1 effector from the blight pathogen, *P. infestans* (PITG-15278) suppress the function of the NRC4-dependent sensor NLR, Rpi-blb2. The cyst nematode effector, SS15, binds to the NB-ARC domain of both inactive and activated forms of NRC2 and NRC3. By binding the NB-ARC domain of NRC2 and NRC3, SS15 is able to suppress their function. The *P. infestans* effector, AVRcap1b, suppresses the function of NRC2 and NRC3, and in the case of NRC3, AVRcap1b suppression requires the ESCRT-related protein NbTOL9a. ESCRT, endosomal sorting complex required for transport; ETI/NTI, effector/NLR-triggered immunity; NB-ARC, nucleotide-binding domain shared with APAF-1, various R-proteins, and CED-4; NLR, nucleotide-binding domain and leucine-rich repeat; NRC, NLR required for cell death.

Our finding that 2 distantly related pathogens, an oomycete and a cyst nematode, evolved effectors to suppress NRC signalling through distinct mechanisms, supports this hypothesis. It is tempting to speculate that NRC redundancy has, therefore, emerged as a strategy to enhance the plant's capacity to evade immune suppression. For example, although both SS15 and AVRcap1b are robust suppressors of NRC2 and NRC3, they are not able to suppress their paralog NRC4. This redundancy would allow the host to mount an effective immune response even in the presence of NRC2 and NRC3 suppressors. However, it should be noted that while we did observe binding between SS15 and NRC4 in Y2H screens, their association was very weak or not detectable *in planta*. It is possible that physiological conditions in the plant do not favour association between SS15 and NRC4, which is consistent with our observations that SS15 cannot suppress NRC4 signalling. Interestingly, the majority of NRC-dependent potato blight R genes, e.g., R1, R8, and Rpi-blb2, signal through NRC4 [24]. NRC4 may therefore have evolved as the helper NLR that predominantly functions with late blight R genes because it evades suppression by *P. infestans*. However, Rpi-amr1 was recently shown to require NRC2 and NRC3 for resistance to *P. infestans*, indicating that the interactions between sensor NLRs, helper NRCs, and pathogen effectors are extremely complex [56]. Nonetheless, this work supports the hypothesis that NLR networks make plant immune systems more resilient through redundant signalling architectures. The extent to which pathogen effectors target NLR networks and the molecular mechanisms by which they do so, however, are still not fully understood.

An emerging paradigm in plant immunity is that NTI and PTI signalling employ common modules and reinforce each other, blurring the division between these 2 classes of plant immunity [57–60]. It is possible that NTI suppressing effectors, such as AVRcap1b and SS15, simultaneously compromise both NTI and PTI by targeting shared immune signalling nodes.

Therefore, identification of NTI suppressors may translate into new insights regarding basal resistance and help us further decipher the intricate nature of the plant immune system.

While the precise molecular mechanism that the cyst nematode effector SS15 utilises to suppress NRC2 and NRC3 remains to be determined, our experiments provide some important insights. SS15 binds both inactive (Figs 6B and S9 and S10) and constitutively active forms of NRCs (Fig 8), presumably through their NB-ARC domain (Fig 7). There are only a few examples of effectors that directly bind NLRs to suppress their activities. NleA, an effector from human enteropathogenic *E. coli* was shown to suppress the NOD-like receptor (NLR) NLRP3 by directly binding to it, thereby interfering with deubiquitination, which is critical for inflammasome activation [61]. NleA associates with both ubiquitinated and nonubiquitinated NLRP3 by interacting with the PYD and LRR domains [62]. While the exact mechanism utilised by NleA is currently unknown, the authors theorise that binding of NleA to the PYD and LRR domains may be responsible for inhibition of NLRP3 inflammasome formation by preventing NLRP3 interaction with other downstream signalling partners or blocking access of the deubiquitinating enzyme to the polyubiquitinated NLR [62]. *P. infestans* IPI-O4 is a plant pathogen effector reported to bind NLRs to suppress host immunity. Chen and colleagues [63] and Karki and colleagues [64] showed that the *P. infestans* effector IPI-O4 compromises the HR mediated by the NLR disease resistance protein RB (also known as Rpi-blb1) by directly binding to its N-terminal CC domain, possibly to compete with binding by the AVR effector AVRblb1, a homolog of IPI-O4. Therefore, IPI-O4 acts directly on the sensor NLR and is more reminiscent of the 3 Rpi-blb2 suppressors we report here than to the NRC suppressors AVRcap1b and SS15 (Fig 13).

The recent elucidation of the ZAR1, RPP1, and ROQ1 structures have revealed that structural remodelling of the NB-ARC domain, a region involved in NLR activation, is essential for resistosome formation [27–30]. The mechanism utilised by SS15 to suppress NRC-mediated immunity may involve tampering with NB-ARC structural rearrangements. In mammalian systems, for example, a compound known as MCC950 directly binds both the inactive and activated forms of NLRP3 to inhibit its function [65,66]. MCC950 binds the central NACHT domain, the mammalian equivalent of the NB-ARC domain of plant NLRs, to interfere with ATP hydrolysis and prevent conformational changes that are critical for NLRP3 activation and subsequent inflammasome assembly. This ultimately drives NLRP3 towards a closed and inactive conformation [65,66]. Based on our findings, we propose that SS15 could be acting as an NLR inhibitor that directly perturbs NRC activities by binding the NB-ARC domain and forcing NRC2 and NRC3 into an inactivated state, possibly through mechanisms analogous to MCC950. Further studies investigating the extent to which SS15 perturbs structural remodelling of the NB-ARC domain of NRCs will provide mechanistic insights into how this effector is able to suppress NRC2 and NRC3.

Unlike SS15, AVRcap1b from the potato late blight pathogen *P. infestans* indirectly suppresses the function of autoimmune NRC2 and NRC3 and therefore likely targets host proteins downstream of these cell death executor NLRs. We identified NbTOL9a, a member of the TOL protein family, as a host target of AVRcap1b and showed that this host protein acts as a negative modulator of NRC3-mediated hypersensitive cell death. TOL proteins are key components of the ESCRT machinery and have well characterised roles in intracellular protein trafficking [52,67,68]. They act as ubiquitin receptors in the early steps of the ESCRT trafficking pathway by interacting with ubiquitinated cargo via their ENTH/VHS and GAT domains [51–53]. Conlan and colleagues [69] identified a TOL family member that is proximal to the *P. syringae* effector AvrPto, when this effector protein was transiently expressed in *N. benthamiana* leaf tissue. In addition, silencing of this TOL protein resulted in decreased growth of *P. syringae* pv. *tabaci* on *N. benthamiana*, which is in line with our observation that TOL proteins

can act as negative regulators of plant immune signalling. The fact that AvrPto is proximal to TOLs further links the NRC network to TOL proteins, since AvrPto is recognised by the NRC2- and NRC3-dependent sensor NLR Prf. Together, these findings strengthen our hypothesis that TOLs can act as negative regulators of this complex immune signalling network. The precise mechanism TOLs utilise to modulate plant immunity, however, is still unknown. In the case of NbTOL9a, this negative regulation does not seem to involve protein–protein interactions between NbTOL9a and NRCs. In mammalian systems, the ESCRT pathway is involved in negatively regulating several forms of programmed cell death, including necroptosis and pyroptosis, by repairing damaged sections of the plasma membrane [70–72]. It is possible that AVRcap1b is co-opting NbTOL9a to hijack a similar immunomodulatory trafficking pathway in the plant cell to counteract NRC-mediated HR cell death and suppress immunity. This model would be consistent with the observation that vesicle trafficking is massively reprogrammed by *P. infestans* effectors during infection [49,73–75].

Interestingly, even though AVRcap1b and SS15 are robust NTI suppressors, they can activate immunity on certain Solanaceae species. AVRcap1b, for example, can activate immunity on accessions of *Solanum capsicibaccatum* carrying the NLR disease resistance gene Rpi-cap1—hence the moniker AVR [32,76]. SS15, on the other hand, is recognised by a yet to be described protein in *Nicotiana tabacum* resulting in HR cell death [35]. The fact that both AVRcap1b and SS15 act as both triggers and suppressors of NLR immunity further highlights the complex coevolutionary dynamics that exist between effectors and NLRs/NLR networks and the need for studies that take into account these intricate epistatic interactions. Additionally, understanding the interplay between AVRcap1b, TOLs, and NRCs will help advance our knowledge of the regulatory mechanisms that govern plant immunity and determine the outcome of multipartite host–pathogen interactions mediated by a complex immune receptor network.

The field of NLR biology has seen significant progress over the past few years, and yet the genetic components and immune signalling pathways downstream of NLR activation remain obscure [77]. Moreover, the precise molecular mechanisms that underpin the activation of paired and networked NLRs and subsequent cell death response are still unknown. Here, we have gained insights into the molecular strategies that plant pathogens utilise to counteract host immune function. We identified 5 effectors (SS15, AVRcap1b, PITG-15278, SS10, and SS34) that compromise components of the NRC network. We focused on SS15 and AVRcap1b, as these effectors counteract NRCs—MADA-type CC-NLRs—which form central nodes within the immune receptor network. SS15 and AVRcap1b, therefore, have the unique potential to uncover valuable mechanistic details regarding the activation of MADA-type CC-NLR resistosomes and their downstream signalling elements. The fact that these distantly related pathogens, an oomycete and a cyst nematode, have independently evolved effectors to counteract NRC2 and NRC3 further highlights the critical role NRCs play in mediating immunity to solanaceous parasites. Beyond SS15 and AVRcap1, there is still much we can learn by studying PTIG-15278, SS10, and SS34, the 3 Rpi-blb2 suppressors identified in this study. Further research may allow us to determine the target(s) and mechanism(s) utilised by these 3 effectors to suppress host immunity. Recently, effectors from the oomycete pathogen *Phytophthora capsici* and the aphid pest *M. persicae* were shown to converge on host E3 SUMO ligase SIZ1 to suppress plant immunity [78]. Taken together with our work, this suggests that microbial pathogens, herbivorous insects, and parasitic nematodes may share more common virulence mechanisms than anticipated. Studying immunosuppressors holds the potential to advance our understanding of the functional principles and evolutionary dynamics that underpin plant immune receptor networks. This knowledge can then be leveraged to guide new approaches for breeding disease resistance to maximise crop protection, for example, by engineering NLRs that evade pathogen suppression.

## Materials and methods

### Plant growth conditions

Wild-type and Rx-transgenic [79] *N benthamiana* lines were grown in a controlled growth chamber with temperature 22 to 25°C, humidity 45% to 65%, and 16/8-hour light/dark cycle.

### Gene cloning and synthesis

**Effector library.** All sequence information for pathogen effectors used in suppression assays can be found in S1 Table.

*P. infestans* effectors, without signal peptide [33,34], were codon optimised for *Solanum tuberosum* and synthesised by GENEWIZ (South Plainfield, NJ, United States of America) into pUC57-Amp. Effector sequences were amplified from the pUC57-Amp vector by Phusion High-Fidelity DNA Polymerase (Thermo Fisher Scientific, MA, USA), and the purified amplicon was used directly for Golden Gate assembly into the Level 0 Universal Acceptor pUAP1 (The Sainsbury Laboratory (TSL) SynBio, addgene no. 63674). Primers used for PCR amplification are listed in S6 Table. Expression constructs were generated by Golden Gate assembly of the Level 0 module into binary vector pICSL86977OD [with cauliflower mosaic virus (CaMV) 35S promoter and octopine synthase gene) terminator] (addgene no. 86180). PITG-05910 and PITG-22926 were synthesised by GENEWIZ (South Plainfield, NJ, USA) into pUC57-kan as Golden Gate L0 modules, then subcloned into binary vector pICH47742 [addgene no. 48001], together with pICH51266 [35S promoter+Ω promoter, addgene no. 50267], and pICH41432 [octopine synthase terminator, addgene no. 50343]. All constructs were verified by sequencing.

*G. rostochiensis* effectors SS4, SS8, SS9, SS15, SS16, and SS19 cloned into pBINPLUS expression vector with N-terminal 4xHA tag were provided by Aska Goverse (Laboratory of Nematology, Wageningen University, the Netherlands) [36,80]. *G. pallida* effector sequences 12N3, 33H17 [81] were synthesised by GENEWIZ (South Plainfield, NJ, USA) as Golden Gate Level 0 modules into pICH41155, Gpa-SS37 [82] was synthesised by GENEWIZ (South Plainfield, NJ, USA) as a Golden Gate Level 0 module into pUC57-kan. The remaining *G. rostochiensis* effector sequences were extracted from the https://parasite.wormbase.org website [37] and synthesised by GENEWIZ (South Plainfield, NJ, USA) as Golden Gate Level 0 modules into pUC57-kan. These effectors were subcloned into binary vector pICH47732 [addgene no. 50434], together with pICH51266 [35S promoter+Ω promoter, addgene no. 50267], and pICH41432 [octopine synthase terminator, addgene no. 50343] for cell death suppression assays.

The green peach aphid (*M. persicae*) and pea aphid (*A. pisum*) effectors cloned into pCB302-3 in *A. tumefaciens* strain GV3101::pM90 were provided by Saskia Hogenhout (John Innes Centre, Norwich, United Kingdom). The tomato bacterial speck pathogen (*P. syringae*) effectors cloned into pEarly Gate 100 in *A. tumefaciens* strain C58C1 were kindly provided by Wenbo Ma (previously University California, Riverside, USA, currently The Sainsbury Laboratory, Norwich, UK).

**Synthesis of NRC constructs.** Full-length *N. benthamiana* NRC2a$^{syn}$ (NCBI accession number KT936525) [24], NRC3$^{WT}$, and NRC4$^{WT}$ were synthesised by GENEWIZ (South Plainfield, NJ, USA) into pICH41155 as Golden Gate L0 modules (S7 Table). These sequences were manually domesticated to remove BpiI and BsaI sites.

**Site-directed mutagenesis of NRC2, NRC3, and SW5b.** Autoactive mutants of NRC2, NRC3, and SW5b were generated by introducing a histidine (H) to arginine (R) mutation in the MHD motif of NRC2 and an aspartic acid (D) to valine (V) substitution in the MHD motif

of NRC3 and SW5b independently. Primers listed in S8 Table were used for introducing mutations by inverse PCR with Phusion High-Fidelity DNA Polymerase (Thermo Fisher Scientific). pICH41155::NRC2, PCR8::NRC3$^{WT}$ [26], and pICH441155::SW5b [24] were used as templates for site-directed mutagenesis of NRC2, NRC3, and SW5b, respectively. The mutated variants were verified by sequencing and then subcloned into pICSL86977OD [addgene no. 86180] (for NRC2$^{H480R}$ and SW5b$^{D847V}$) and pICH86988 [addgene no. 48076] (NRC3$^{D480V}$). Autoactive NRC4 mutants used in this study were described previously [25]. Verified plasmids were then transformed into GV3101:pM90 for cell death assay screens.

To determine whether an intact p-loop is essential for SS15 *in planta* association to NRC2 and NRC3, a lysine (K) to arginine (R) mutation was introduced independently into the p-loops of both proteins by site-directed mutagenesis using Phusion High-Fidelity DNA Polymerase (Thermo Fisher Scientific). pCR8::NRC2$^{WT}$-ns and PCR8::NRC3$^{WT}$-ns [26] were used as a templates. Primers used to for introducing mutations are listed in S8 Table. The mutated NRC variants were verified by sequencing and subcloned into pICH86988 [addgene no. 48076] together with pICSL50010 [C-terminal 4xMyc tag, addgene no. 50310]. Both constructs were verified by sequencing and transformed into *A. tumefaciens* GV3101::pM90.

**Cloning into *E. coli* expression vectors for protein purification.** SS15: DNA encoding SS15 residues Ser-25 to Ile-246 (lacking signal peptide) was amplified from pBINPLUS::4HA::SS15 plasmid (using primers shown in S15 Table) and cloned into the pOPINS3C vector resulting in an N-terminal 6xHis-SUMO tag with SS15, linked by a 3C cleavage site (pOPINS3C:SS15) [83].

NB-ARC domains of NRC2, NRC3, and NRC4: For NRC2, 2 NB-ARC domain constructs with alternative domain boundaries were generated. DNA encoding NRC2 NB-ARC domain residues Val-148 to Tyr-508 (NRC2$^{148-508}$) or Val-148 to Asn-496 (NRC2$^{148-496}$) was amplified from pICH41155::NRC2a$^{syn}$ (using primers shown in S15 Table) and cloned into the vector pOPINS3C resulting in an N-terminal 6xHis-SUMO tag with NRC2 NB-ARC, linked by a 3C cleavage site (pOPINS3C:NRC2$^{148-508}$ and pOPINS3C:NRC2$^{148-496}$, respectively). Similarly, DNA encoding NRC3 NB-ARC residues Val-152 to Ser-507 (NRC3$^{NB-ARC}$) was amplified from pICH41155::NRC3$^{WT}$ (using primers shown in S15 Table) and DNA encoding NRC4 NB-ARC domain Ala-150 to Lys-491 (NRC4$^{NB-ARC}$) was amplified from pICH41155::NRC4$^{WT}$ (using primers shown in S15 Table) and cloned into the vector pOPINS3C as described above (pOPINS3C:NRC3$^{NB-ARC}$ and pOPINS3C:NRC4$^{NB-ARC}$, respectively).

**Generating C-terminally 4xMyc tagged NRC2 and NRC3 autoactive mutants.** To generate NRC2$^{H480R}$::4xMyc and NRC3$^{D480V}$::4xMyc, the stop codon was removed from pICSL86977OD::NRC2$^{H480R}$ and pICH86988::NRC3$^{D480V}$ by Phusion High-Fidelity DNA polymerase (Thermo Fischer Scientific) using primers in S8 Table. The purified amplicon was used directly for Golden Gate assembly into binary vector pICH86988 [addgene no. 48076], together with pICSL50010 [C-terminal 4xMyc tag, addgene no. 50310]. Both constructs were verified by DNA sequencing and then transformed into *A. bacterium* strain GV3101::pM90.

**Generating N-terminally GFP-tagged, C-terminally 6xHA-tagged, and C-terminally 4xmyc-tagged AVRcap1b.** AVRcap1b constructs used in coIP experiments were generated by Golden Gate assembly. To generate GFP::AVRcap1b, level 0 pUAP1::AVRcap1b was assembled into binary vector pICH86966 [addgene no. 48075] driven by pICSL13008 [CaMV 35S promoter+5′ untranslated leader tobacco mosaic virus, TSL SynBio], pICSL30006 [GFP, addgene no. 50303], and pICH41432 [octopine synthase terminator, addgene no. 50343]. To generate AVRcap1b::6xHA and AVRcap1b:L4xmyc, the stop codon was removed from pUAP1::AVRcap1b by Phusion High-Fidelity DNA Polymerase (Thermo Fisher Scientific) using primers listed in S6 Table. The purified amplicon was used directly for Golden Gate assembly into binary vector pICH47742 [addgene no. 48001] driven by pICH85281

[mannopine synthase promoter+Ω (MasΩpro), addgene no. 50272], pICSL50009 [6xHA, addgene no. 50309], and pICSL60008 [*Arabidopsis* heat shock protein terminator (HSPter), TSL SynBio]. All constructs were verified by DNA sequencing and then transformed into *A. bacterium* strain GV3101::pM90.

**Cloning of NbTOL paralogs.** We used reciprocal BLAST searches of Nbv6.1trP4561, which was identified in both IP-MS and Y2H assays, and mined the V6.1 transcriptome (https://benthgenome.qut.edu.au/) for additional TOL sequences. We cross-validated the extracted TOLs with the genome-based predicted proteomes [84] and removed likely duplicates representing chimeric transcripts due to assembly of short read sequences. We identified a total of 5 TOL paralogs, which we termed NbTOL9a (Nbv6.1trP4361), NbTOL9b (Nbv6.1trP9166), NbTOL3 (Nbv6.1trA40123), NbTOL6 (Nbv6.1trP73492), and NbTOL9c (Nbv6.1trA64113) (S5 Table). NbTOL paralogs were synthesised using GENEWIZ (South Plainfield, NJ, USA) into pUC57-Kan as L0 modules. Individual TOL paralogs were subjected to PCR by Phusion High-Fidelity DNA Polymerase (Thermo Fisher Scientific) to remove the stop codon (primer information listed in S12 Table). The purified amplicons were used directly for Golden Gate assembly into binary vector pICH47732 [addgene no. 50434], together with pICH51266 [35S promoter+Ω promoter, addgene no. 50267], pICSL50009 [6xHA, addgene no. 50309], and pICH41432 [octopine synthase terminator, addgene no. 50343]. All constructs were verified by DNA sequencing and transformed into *A. tumefaciens* GV3101::pM90.

## Cell death and suppression assays by agroinfiltration

Information of constructs used for cell death assays are summarised in S9 Table. Transient expression of NLR immune receptors and cognate effectors and autoactive immune receptors with EV or effector constructs were performed according methods previously described [13]. Briefly, 4- to 5-week-old *N. benthamiana* plants were infiltrated with *A. tumefaciens* GV3101::pM90 strains carrying the expression vectors of different indicated proteins within the text. *A. tumefaciens* suspensions were adjusted in infiltration buffer (10 mM MES, 10 mM $MgCl_2$, and 150 μM acetosyringone (pH5.6)) to final $OD_{600}$ indicated in S9 Table. $OD_{600}$ for all effectors were adjusted to 0.2. For NbTOL9a overexpression assays, NbTOL9a::6xHA construct was coinfiltrated at a final $OD_{600}$ of 0.2. The cell death, HR, phenotype was scored 5 to 7 days after agroinfiltration, unless otherwise stated, using a previously described scale [85], modified from 0 (no necrosis observed) to 7 (confluent necrosis).

## Virus-induced gene silencing (VIGS) of NRC homologs

VIGS was performed in *N. benthamiana* as previously described [86]. Suspensions of *A. tumefaciens* strain GV3101::pM90 harbouring TRV RNA1 (pYL155) and TRV RNA2 (pYL279), with corresponding fragments from *NRC2*, *NRC3*, and *NRC4*, were mixed in a 2:1 ratio in infiltration buffer (10 mM 2-[N-morpholine]-ethanesulfonic acid [MES]; 10 mM $MgCl_2$; and 150 μM acetosyringone (pH 5.6)) to a final OD600 of 0.3. Two-week-old *N. benthamiana* plants were infiltrated with *A. tumefaciens* for VIGS assays; upper leaves were used 2 to 3 weeks later for further agroinfiltrations. The *NRC2/3* double silencing, *NRC4* and *NRC2/3/4* triple silencing constructs were described previously [24,26].

## PVX infection assays (agroinfection)

Plants were used 3 weeks after TRV infection. Plants were infected with PVX (pGR106) using the toothpick inoculation method, expressed via *A. tumefaciens* [24,46], and examined for the spread of trailing necrotic lesions from the inoculated spots [43]. pGR106::PVX::GFP

construct generation was described previously [24]. One day before PVX toothpick inoculation, EV and pICH86977::AVRcap1b or pBINPLUS::4HA::SS15 constructs were expressed by agroinfiltration into leaves of *Rx* plants that were subjected to *NRC2/3* double, *NRC4* and *NRC2/3/4* triple silencing. The infiltrated area was then circled with a marker pen. *NRC* homologs were silenced by VIGS as described above in *Rx*-transgenic *N. benthamiana*. Toothpicks were dipped into culture of *A. tumefaciens* harbouring the PVX-GFP vector and then used to pierce small holes in the leaves of *N. benthamiana*. Photos were taken at 3 weeks after PVX inoculation, and the size of the lesions were measured in Fiji (formerly ImageJ). Scatterplot of the lesion size was generated in R using the ggplot2 package, as described previously [87]. A core borer (0.8 cm$^2$) was used to collect leaf discs from the inoculation sites 3 weeks after PVX inoculation for immunoblot with anti-GFP (B-2, sc9996 HRP, Santa Cruz Biotechnology, CA, USA). GFP accumulation was visualised with Pierce ECL Western (32106, Thermo Fisher Scientific) and where necessary up to 50% SuperSignal West Femto Maximum Sensitivity Substrate (34095, Thermo Fisher Scientific).

## Phylogenetic analyses of *Nicotiana benthamiana* and *Arabidopsis thaliana* TOL proteins

Amino acid sequences of the NbTOL paralogs identified in *N. benthamiana* and previously published *A. thaliana* AtTOL proteins [53,88] were aligned using Clustal Omega [89]. The alignment was then manually edited in MEGAX [90]. The gaps in the alignment were manually removed, and only the ENTH and GAT domains were used to generate the phylogenetic tree. A maximum-likelihood tree of the *N. benthamiana* and *A. thaliana* TOLs was generated in MEGAX using the JTT model and with bootstrap values based on 1,000 iterations (S12 Fig). The resulting tree was then visualised using iTOL [91]. The alignment used to make the tree is provided as a S13 Data.

## Hairpin RNA-mediated gene silencing

The silencing fragment was amplified out of *N. benthamiana* cDNA by Phusion High-Fidelity DNA Polymerase (Thermo Fisher Scientific) using primers listed in S10 Table. The specificity of the silencing fragment was analysed using the *N. benthamiana* genome sequence and associated gene silencing target prediction tool (SGN VIGS tool: https://vigs.solgenomics.net). The purified amplicon was cloned into pRNAi-GG vector according to Yan and colleagues [92]. Construct was verified by DNA sequencing and then transformed into *A. tumefaciens* strain GV3101::pM90. Silencing evading NbTOL9a (NbTOL9a$^{syn}$) was synthesised by GENEWIZ (South Plainfield, NJ, USA) in pUC57-kan and generated according to Wu and colleagues [24]. Leaves were coinfiltrated with either pRNAi-GG::*NbTOL9a* or pRNAi-GG::*GUS*, at a final OD$_{600}$ of 0.5, together with different proteins indicated in the text with final OD$_{600}$ indicated in S10 Table. The HR cell death on the leaves was scored at 5 to 7 days as described above.

## Protein–protein interaction studies

**Yeast two-hybrid screens.** Unbiased Y2H screens were performed by Hybrigenics Services (http://www.hybrigenics.com, Paris, France). For AVRcap1b (residues Ala62 –Pro678, lacking the signal peptide) was cloned into both the pB27 and pB66 bait plasmids, as C-terminal fusions to LexA (LexA-AVRcap1b) and Gal4 (Gal4-AVRcap1b), respectively. The 2 screens were performed against a randomly primed *N. benthamiana*–mixed tissue cDNA library. For LexA-AVRcap1b, a total of 52.8 million interactions were screened (approximately 5-fold library coverage), and 9 positive clones were fully analysed. For Gal4-AVRcap1b, a total of 87.6

million interactions were screened (approximately 8-fold library coverage), and 26 positive clones were fully analysed. These clones correspond to 14 different annotated proteins, one of which was unknown (S2 Table). For SS15 (residues Ser25 –stop 246, lacking the signal peptide) was cloned into pB27 as a C-terminal LexA bait (LexA-SS15) and screened against a randomly primed *N. benthamiana*–mixed tissue cDNA library. A total of 61.4 million interactions were screened (approximately 6-fold library coverage), and 202 positive clones were fully processed, corresponding to 10 different annotated proteins, 3 of which were unknown (S3 Table). Interactions were categorised based on their Predicted Biological Score, a measure to assess the interaction reliability, which is calculated based on a statistical model of the competition for bait binding between fragments [93,94].

To narrow down the region mediating protein–protein interactions between SS15 and NRCs, we generated NRC2, NRC3, and NRC4 CC and NB-ARC domain truncations and ran a Y2H experiment using the Matchmaker Gold system (Takara Bio, USA). Plasmid DNA encoding AVRcap1b (negative control) and SS15 in pGBKT7 (bait), generated in this study (using primers shown in S11 Table), were cotransformed into chemically competent Y2HGold cells (Takara Bio, USA) with individual NRC truncates in pGADT7 (prey) (using primers shown in S11 Table), as described previously [95,96], in the AH109 yeast strain. Single colonies grown on selection plates were inoculated in 5 ml of SD$^{-Leu-Trp}$ overnight at 28˚C (ST0047, Takara Bio, USA). Saturated culture was then used to make serial dilutions of OD$_{600}$ 1, $10^{-1}$ and $10^{-2}$, respectively. A volume of 3 μl of each dilution was then spotted on a SD$^{-Leu-Trp}$ plate (ST0048, Takara Bio, USA) as a growth control and on a SD$^{-Leu-Trp-Ade-His}$ plate (ST0054, Takara Bio, USA) containing X-α-gal and supplemented with 0.2% Adenine. Plates were incubated for 3 to 6 days at 28˚C and then imaged. Each experiment was repeated a minimum of 3 times, with similar results. The commercial yeast constructs were used as positive (pGBKT7-53/pGADT7-T) and negative (pGBKT7-Lam/pGADT7-T) controls (Clontech, Takara Bio, CA, USA).

***In planta* CoIPs.** Protein samples were extracted from 2 *N. benthamiana* leaves 3 days post-agroinfiltration and homogenised in GTEN extraction buffer [10% glycerol, 25 mM Tris-HCl (pH 7.5), 1 mM EDTA, 150 mM NaCl, 2% (w/v) polyvinylpolypyrrolidone, 10 mM dithiothreitol, 1× protease inhibitor cocktail (SIGMA), 0.2% IGEPAL (SIGMA, United Kingdom)] [97]. The final concentration OD$_{600}$ used for each protein is indicated in S13 Table. After centrifugation at 5,000 ×*g* for 20 minutes, the supernatant was passed through a Minisart 0.45 μM filter (Sartorius Stedim Biotech, Goettingen, Germany) and used for SDS-PAGE. For coIP, 1.4 ml of filtered total protein extract was mixed with 30 μl of GFP-Trap-A agarose beads (Chromatek, Munich, Germany), anti-c-myc agarose beads (A7470, SIGMA) or anti-HA affinity matrix beads (Roche, Switzerland) and incubated end over end for 1 hour at 4˚C. Beads were washed 5 times with immunoprecipitation wash buffer [GTEN extraction buffer with 0.3% (v/v) IGEPAL (SIGMA)] and resuspended in 70 μl SDS loading dye. Proteins were eluted from beads by heating at 10 minutes at 70˚C (for GFP) or 95˚C (for myc or HA). Immunoprecipitated samples were separated by SDS-PAGE and transferred onto a polyvinylidene difluoride membrane using Trans-Blot turbo Transfer system (Bio-Rad, Munich), according to the manufacturer's instructions. Blots were preblocked with 5% skim milk powder in Tris-buffered saline plus Tween 20 (TBS-T) overnight in 4˚C or for a minimum of 1 hour at room temperature. Epitope tags were detected with HA-probe (F-7) horse radish peroxidase (HRP)-conjugated antibody (Santa Cruz Biotechnology), c-Myc (9E10) HRP (Santa Cruz Biotechnology), or anti-GFP (B-2) HRP (Santa Cruz Biotechnology) antibody in a 1:5,000 dilution in 5% skim milk powder in TBS-T. Proteins were visualised with Pierce ECL Western (32106, Thermo Fisher Scientific) and where necessary up to 50% SuperSignal West Femto Maximum Sensitivity Substrate (34095, Thermo Fisher Scientific). Membrane imaging was carried out

with an ImageQuant LAS 4000 luminescent imager (GE Healthcare Life Sciences, Piscataway, NJ). Rubisco loading control was stained using Pierce (24580, Thermo Fisher Scientific) or Instant Blue (Expedeon, Cambridge, United Kingdom).

**Protein purification from *E. coli* and in vitro protein–protein interaction studies.** Recombinant SS15 protein (lacking signal peptide) was produced using *E. coli* SHuffle cells [98] transformed with pOPINS3C:SS15 (see Gene cloning and synthesis section above). Cell culture was grown in autoinduction media [99] at 30°C to an $A_{600}$ 0.6 to 0.8 followed by overnight incubation at 18°C and harvested by centrifugation. Pelleted cells were resuspended in 50 mM Tris HCl (pH 8), 500 mM NaCl, 50 mM Glycine, 5% (vol/vol) glycerol, and 20 mM imidazole (buffer A) supplemented with cOmplete EDTA-free protease inhibitor tablets (Roche) and lysed by sonication. The clarified cell lysate was applied to a $Ni^{2+}$-NTA column connected to an AKTA pure system. 6xHis+SUMO-SS15 was step eluted with elution buffer (buffer A containing 500 mM imidazole) and directly injected onto a Superdex 200 26/600 gel filtration column preequilibrated in buffer B (20 mM HEPES (pH 7.5), 150 mM NaCl, and 1 mM TCEP). The fractions containing 6xHis+SUMO-SS15 were pooled and concentrated to 2 to 3 mg/ml. The 6xHis+SUMO tag was cleaved by addition of 3C protease (10 μg/mg fusion protein) and incubation overnight at 4°C. Cleaved SS15 was further purified using a $Ni^{2+}$-NTA column (collecting the eluate) followed by gel filtration as above. The concentration of protein was judged by absorbance at 280 nm (using a calculated molar extinction coefficient of SS15, 35920 $M^{-1}cm^{-1}$).

Recombinant $NRC3^{NB-ARC}$ was produced using *E. coli* Lemo21 (DE3) cells transformed with pOPINS3C:$NRC3^{NB-ARC}$ (see Gene cloning and synthesis section for details). Cell culture was grown in autoinduction media at 30°C to an $A_{600}$ 0.6 to 0.8 followed by overnight incubation at 18°C and harvested by centrifugation. Pelleted cells were resuspended in buffer A supplemented with EDTA-free protease inhibitor tablets and lysed by sonication. The clarified cell lysate was applied to a $Ni^{2+}$-NTA column connected to an AKTA pure system. 6xHis+-SUMO-$NRC3^{NB-ARC}$ was step eluted with elution buffer (buffer A containing 500 mM imidazole) and directly injected onto a Superdex 200 26/600 gel filtration column preequilibrated in buffer B (20 mM HEPES (pH 7.5), 150 mM NaCl, and 1 mM TCEP). The fractions containing 6xHis+SUMO- $NRC3^{NB-ARC}$ were pooled and concentrated to 1 mg/ml. The 6xHis+SUMO tag was cleaved by addition of 3C protease (10 μg/mg fusion protein) and incubation overnight at 4°C. Cleaved $NRC3^{NB-ARC}$ was further purified using a $Ni^{2+}$-NTA column (collecting the eluate) followed by gel filtration as above. The concentration of protein was judged by absorbance at 280 nm (using a calculated molar extinction coefficient of $NRC3^{NB-ARC}$, 48020 $M^{-1}cm^{-1}$).

To purify SS15 in complex with $NRC3^{NB-ARC}$, *E. coli* cultures were grown individually for SS15 and $NRC3^{NB-ARC}$ and harvested by centrifugation as explained above. Pelleted cells expressing SS15 and $NRC3^{NB-ARC}$ were resuspended separately in buffer A supplemented with EDTA-free protease inhibitor tablets. After resuspension, SS15 and $NRC3^{NB-ARC}$ expressing cells were mixed and lysed together by sonication. The clarified cell lysate (containing 6xHis+-SUMO-$NRC3^{NB-ARC}$ and 6xHis-SUMO-SS15) was applied to a $Ni^{2+}$-NTA column connected to an AKTA pure system. 6xHis+SUMO-tagged protein (SS15 and $NRC3^{NB-ARC}$) was step eluted with elution buffer (buffer A containing 500 mM imidazole) and directly injected onto a Superdex 200 26/600 gel filtration column preequilibrated in buffer B (20 mM HEPES (pH 7.5), 150 mM NaCl, and 1 mM TCEP). The fractions containing 6xHis+SUMO-SS15 and 6xHis+SUMO-$NRC3^{NB-ARC}$ were pooled and concentrated to 1 to 2 mg/ml. The 6xHis +SUMO tag was cleaved by addition of 3C protease (10 μg/mg fusion protein) and incubation overnight at 4°C. Cleaved $NRC3^{NB-ARC}$ and SS15 complex was further purified using a $Ni^{2+}$-NTA column (collecting the eluate) followed by gel filtration as above. The fractions

containing complex of NRC3$^{NB-ARC}$ and SS15 were pooled together and concentrated. The concentration of complex was judged by absorbance at 280 nm (using a calculated molar extinction coefficient of SS15 and NRC3$^{NB-ARC}$, 83940 M$^{-1}$cm$^{-1}$).

## IP-MS

Total proteins were extracted from *N. benthamiana* leaves 3 days after agroinfiltration of GFP::AVRcap1b or GFP::PexRD54 [74] and subjected to immunoprecipitation using GFP_Trap_A beads (Chromotek, Munich, Germany), as described previously [50,97,100]. PexRD54 was included as a control, as it is also a large *P. infestans* RxLR effector and extensive studies suggests that its role is likely independent of the NRC network [74,75]. Final OD$_{600}$ for each protein is indicated in S13 Table. Immunoprecipitated samples were separated by SDS-PAGE (4% to 20% gradient gel, Biorad, United Kingdom) and stained with Coomassie brilliant Blue G-250 (SimplyBlue Safe stain, Invitrogen). Enriched protein samples were cut out of the gel (approximately 5 × 10 mm) and digested with trypsin. Extracted peptides were analysed by liquid chromatography–tandem mass spectrometry (LC–MS/MS) with the Orbitrap Fusion mass spectrometer and nanoflow-HPLC system U3000 (Thermo Fisher Scientific, UK) [48,50]. A total of 3 biological replicates for each sample type were submitted.

**Mass spectrometry data processing.** As previously described [48–50,97,100], peak lists were extracted from raw data using MS Convert [101] and peptides identified on Mascot server 2.4.1 using Mascot Daemon (Matrix Science). The peak lists were searched against the *N. benthamiana* genome database called Nicotiana_Benthamiana_Nbv6trPAplusSGNU-niq_20170808 (398,682 sequences; 137,880,484 residues), supplemented with common contaminants. The search settings allowed tryptic peptides with up to 2 possible miscleavages and charge states +2, +3, +4, carbamidomethylated Cysteine (static) and oxidised Methionine (variable) modifications and monoisotopic precursor and fragment ion mass tolerances 10 ppm and 0.6 Da, respectively. A decoy database was used to validate peptide sequence matches. The Mascot results were combined in Scaffold 4.4.0 (Proteome Software), where thresholds for peptide sequence match and inferred protein were set to exceeded 95.0% and 99%, respectively, with at least 2 unique peptides identified for each protein. The aforementioned protein list with spectral counts as quantitative values was exported and further analysed in Excel (Microsoft). More information on peptides identified is available in S14 Table.

The MS proteomics data have been deposited to the ProteomeXchange Consortium via the PRIDE [102] partner repository with the dataset identifier PXD023178 and 10.6019/PXD023178.

## Supporting information

**S1 Fig. Effectors can suppress ETI/NTI.** (A) ETI/NTI suppressing effectors counteract the activation of NLR immunity elicited by other effectors with an AVR activity (AVR effectors). The majority of effectors studied to date are known to suppress PTI by acting on various host targets involved in PTI signalling. However, we know little about the mechanisms of NTI suppressing effectors and how these effectors counteract NLR receptor functions and/or resistosome formation (B) The NRC network of Solanaceae. Helper NLRs (NRC2, NRC3, and NRC4) function redundantly with a series of R genes that confer resistance against multiple pathogens. These R genes encode NLR sensors that have specialised in detecting effectors from pathogens as diverse as oomycetes, bacteria, nematodes, viruses, and aphids. Many of the NRC-dependent R genes are agronomically important. Sensor NLRs (blue) with N-terminal extensions possess an additional domain at the N-terminus, represented in dark blue. The extent to which pathogen effectors have evolved to target the NRC helpers are currently unknown. Figure is adapted from [24,55]. AVR, avirulence; ETI/NTI, effector/NLR-triggered

immunity; NLR, nucleotide-binding domain and leucine-rich repeat; NRC, NLR required for cell death; PAMP, pathogen-associated molecular pattern; PRR, pattern recognition receptor; PTI, PRR-triggered immunity.
(TIF)

**S2 Fig. Statistical analysis of cell death suppression assay for 5 candidate effectors coinfiltrated with Pto/AvrPto or Rpi-blb2/AVRblb2.** Statistical analysis was conducted using besthr R library (MacLean, 2019) (S4 Data). (A–D) Each panel corresponds to results from Pto/AvrPto or Rpi-blb2/AVRblb2 (labelled above), coexpressed with either EV (dark blue), AVRcap1b (green), HopAB2 (dark purple), PITG-15278 (light purple), SS10 (light blue), SS15 (dark orange), or SS34 (light orange), where EV was used as a negative control, labelled below plots. The left panel represents the ranked data (dots) and their corresponding means (dashed lines), with the size of each dot proportional to the number of observations for each specific value (count key below each panel). We performed bootstrap resampling tests using a lower significance cutoff of 0.025 and an upper cutoff of 0.975. Mean ranks of test samples falling outside of these cutoffs in the control samples bootstrap population were considered significant. The panels on the right shows the distribution of 1,000 bootstrap sample rank means, where the blue areas under the curve illustrate the 0.025 and 0.975 percentiles of the distribution. EV, empty vector.
(TIF)

**S3 Fig. HopAB2 suppression of Prf-mediated cell death in *N. benthamiana* is not evident in young leaves.** *P. syringae* effectors HopA1, HopAB1, HopAB2, HopAB3, and HopAF1 were transiently coexpressed with Pto/AvrPto in *N. benthamiana* leaves. The leaves were photographed 5 days after infiltration. HopAB2 suppression of Pto/AvrPto in *N. benthamiana* were evident on leaf 5 (old leaf, left), but not on leaf 3 (young leaf, right) as defined by [39]. Photographs are representative of 3 technical repeats. EV, empty vector.
(TIF)

**S4 Fig. AVRcap1b and SS15 suppress cell death activity mediated by autoactive NRC2$^{H480R}$ and NRC3$^{D480V}$, but not of NRC4$^{D478V}$ mutants.** (A–F) Leaf panels: Photo of representative *N. benthamiana* leaves showing HR after coexpression of autoimmune NRC2$^{H480R}$, NRC3$^{D480V}$, and NRC4$^{D478V}$ mutants, indicated to the left of leaf panels, with either EV, AVRcap1b, PITG_15278, HopAB2, SS10, SS15, or SS34 (labelled on leaf) (S4 Data). Middle panels: HR was scored 5 days post-agroinfiltration, using a modified 0–7 scale (Segretin and colleagues) and photographed at 5 days after agroinfiltration. HR results are presented as dot plots, where the size of each dot is proportional to the number of samples with the same score (count) within each biological replicate. Each effector is represented by a different dot colour (see key right side of plot). The experiment was independently repeated 3 times each with 6 technical replicates. The columns of each tested effector correspond to results from different biological replicates. Right panels: Statistical analysis was conducted using besthr R library (MacLean, 2019). The dots represent the ranked data and their corresponding means (dashed lines), with the size of each dot proportional to the number of observations for each specific value (count key below each panel). The panels on the right show the distribution of 1,000 bootstrap sample rank means, where the blue areas under the curve illustrates the 0.025 and 0.975 percentiles of the distribution. A difference is considered significant if the ranked mean for a given condition falls within or beyond the blue percentile of the mean distribution of the EV control. Significant differences between the conditions are indicated with an asterisk (*). EV, empty vector; HR, hypersensitive response.
(TIF)

**S5 Fig. Statistical analysis of cell death suppression assay for AVRcap1b and SS15 coexpressed with autoactive NRC2$^{H480R}$, NRC3$^{D480V}$, and NRC4$^{D478V}$.** Statistical analysis was conducted using besthr R library (MacLean, 2019) (S5 Data). Each panel corresponds to results from (A, D) NRC2$^{H480R}$, (B, E) NRC3$^{D480V}$, or (C, F) NRC4$^{D478V}$ (labelled above), coexpressed with either EV (blue), AVRcap1b (green), or SS15 (orange), where EV was used as a negative control (labelled under plots). The left panel represents the ranked data (dots) and their corresponding means (dashed lines), with the size of each dot proportional to the number of observations for each specific value (count key below each panel). The panels on the right show the distribution of 1,000 bootstrap sample rank means, where the blue areas under the curve illustrate the 0.025 and 0.975 percentiles of the distribution. A difference is considered significant if the ranked mean for a given condition falls within or beyond the blue percentile of the mean distribution of the EV control. EV, empty vector.
(TIF)

**S6 Fig. Statistical analysis of suppression of NRC2/3-dependent sensor NLRs by AVRcap1b or SS15.** Statistical analysis was conducted using besthr R library (MacLean, 2019) (S6 Data). (A–D) Within each panel results from a range of different sensor NLRs (labelled below) were coexpressed with either EV (blue), AVRcap1b (green), or SS15 (orange), where EV was used as a negative control (see key on right hand side). Panels (A) and (C) represent the ranked data (dots) and their corresponding means (dashed lines), with the size of each dot proportional to the number of observations for each specific value (refer to count key, on left hand side). Panels (B) and (D) show the distribution of 1,000 bootstrap sample rank means, where the blue areas under the curve illustrate the 0.025 and 0.975 percentiles of the distribution. A difference is considered significant if the ranked mean for a given condition falls within or beyond the blue percentile of the mean distribution of EV control for each treatment. EV, empty vector; NLR, nucleotide-binding domain and leucine-rich repeat.
(TIF)

**S7 Fig. Statistical analysis shows that AVRcap1b and SS15 suppress Rx-mediated cell death in NRC4-silenced Rx-transgenic *N. benthamiana* plants.** Statistical analysis was conducted using besthr R library (MacLean, 2019) (S7 Data). (A–F) Each panel represents a different silencing treatment (labelled above). Left plot represents the ranked data (dots) and their corresponding means (dashed lines), with the size of each dot proportional to the number of observations for each specific value (refer to count key below each individual panel). Right plot shows the distribution of 1,000 bootstrap sample rank means, where the blue areas under the curve illustrate the 0.025 and 0.975 percentiles of the distribution. A difference is considered significant if the ranked mean for a given condition falls within or beyond the blue percentile of the mean distribution of EV control. EV, empty vector.
(TIF)

**S8 Fig. SS15 binds the NB-ARC domain of NRC proteins in yeast.** Y2H assay of SS15 and AVRcap1b with CC and NB-ARC domains of NRC2, NRC3, and NRC4. Control plates for yeast growth are on the left (SD-Leu-Trp, ST0048, Takara Bio, USA), with quadruple dropout media supplemented with X-a-gal on the right (SD-Leu-Trp-Ade-His, ST0054, Takara Bio, USA). The commercial yeast constructs were used as positive (pGBKT7-53/pGADT7-T) and negative (pGBKT7-Lam/pGADT7-T) controls. Growth and development of blue colouration in the selection plate are both indicative of protein–protein interaction. SS15 and AVRcap1b were fused to the GAL4 DNA-binding domain (Bait), and the truncated NRCs were fused to the GAL4-activator domain (Prey). Each experiment was repeated 3 times with similar results. CC, coiled-coil; NB-ARC, nucleotide-binding domain shared with APAF-1, various R-

proteins, and CED-4; NRC, NLR required for cell death; Y2H, yeast two-hybrid.
(TIF)

**S9 Fig. CoIP experiment between SS15 and NRC2, NRC3, and NRC4.** N-terminally 4xHA-tagged SS15 (SS15) and SS9 (SS9) were individually transiently coexpressed with NRC proteins fused to a C-terminal 4xMyc tag. SS9 was used as a negative control. IP were performed with agarose beads conjugated to either anti-Myc (MYC IP) or anti-HA (HA IP) antibodies. Total protein extracts were immunoblotted with appropriate antisera labelled on the left. Approximate molecular weights (kDa) of the proteins are shown on the right. Rubisco loading controls were conducted using CBB staining. This experiment is a representative of 2 independent replicates. CBB, Coomassie brilliant blue; coIP, co-immunoprecipitation; IP, immunoprecipitation.
(TIF)

**S10 Fig. SS15 associates with NRC2, NRC3, and NRC4 p-loop mutants in planta.** (A) N-terminally 4xHA-tagged SS15 was transiently coexpressed in *N. benthamiana* with C-terminally 4xMyc-tagged NRC2, NRC4, NRC2$^{K188R}$, NRC3$^{K191R}$, and NRC4$^{K190R}$. IP was performed with agarose beads conjugated to anti-Myc (Myc IP) antibodies. Total protein extracts (Input) and proteins obtained by coIP were immunoblotted with appropriate antisera labelled on the right. Approximate molecular weights (kDa) of the proteins are shown on the right. Rubisco loading controls were conducted using CBB staining. This experiment is representative of 3 independent replicates. CBB, Coomassie brilliant blue; coIP, co-immunoprecipitation; IP, immunoprecipitation.
(TIF)

**S11 Fig. Target of AVRcap1b identified via IP-MS.** (A) Immune affinity enriched protein abundance plot generated based on proteins groups identified for AVRcap1b and PexRD54. Protein abundances were calculated as the total mean spectral counts identified with each effector across 3 replicates. The Log$_2$ of the ratio of mean counts observed for AVRcap1b divided by the mean counts observed for PexRD54 was calculated. All 261 proteins identified were then sorted from highest to lowest ratio and plotted in this order (see S14 Table for details). Proteins with the highest ratio correspond to the most likely AVRcap1b (blue) specific interactors. Proteins most likely to correspond to PexRD54 are labelled in green. Proteins in grey were identified in both effectors. (B) Eight putative unique AVRcap1b proteins (based on previously mentioned ratio of mean spectral counts) identified in IP-MS experiment. Unique spectral counts obtained in each of the 3 independent replicates with AVRcap1b are included on the right-hand side. IP-MS, immunoprecipitation–mass spetrometry.
(TIF)

**S12 Fig. Maximum-likelihood phylogenetic tree of TOL proteins from *N. benthamiana* and *A. thaliana*.** Protein sequences were aligned using Clustal Omega. The ENTH and GAT domains were used for further analysis. The phylogenetic tree was constructed in MEGAX using the JTT substitution model and 1,000 bootstrap iterations. Branches with bootstrap support higher than 80 are indicated with red dots. NbTOL9a is indicated by 2 red asterisks. The scale bar indicates the evolutionary distance in amino acid substitutions per site. JTT, Jones–Taylor–Thornton; TOL, Target of Myb 1-like protein.
(TIF)

**S13 Fig. RNAi::*NbTOL9a* silencing construct effectively reduces protein accumulation levels of NbTOL9a.** WT NbTOL9a (NbTOL9a::6HA) and the synthetic version (synNbTOL9a::6xHA) were transiently coexpressed with RNAi::*GUS* or RNAi:*NbTOL9a*.

synNbTOL9a::6xHA was used as a control that is not targeted for knockdown by the RNAi::
*NbTOL9a*. Total protein extracts were immunoblotted with HA antiserum. Approximate molecular weights (kDa) of the proteins are shown on the right. Rubisco loading controls were conducted using Pierce staining. WT, wild type.
(TIF)

**S14 Fig. Statistical analysis showing silencing of NbTOL9a enhanced cell death mediated by NRC2$^{H480R}$ and NRC3$^{D480V}$ but not ZAR1$^{D481V}$ or NRC4$^{D478V}$.** Statistical analysis was conducted using besthr R library (MacLean, 2019) (S10 Data). Each panel represents increasing concertation of *A. tumefaciens* expressing NRC3$^{D480V}$ and NRC4$^{D478V}$ (OD$_{600}$ = 0.1, 0.25, or 0.5). (A) NRC2$^{H480R}$, (B) NRC3$^{D480V}$, (C) NRC4$^{D478V}$, and (D) ZAR1$^{D481V}$. The left most panel represents the ranked data (dots) and their corresponding means (dashed lines), with the size of each dot proportional to the number of observations for each specific value (refer to count legend bottom of figure). The right panel shows the distribution of 1,000 bootstrap sample rank means, where the blue areas under the curve illustrate the 0.025 and 0.975 percentiles of the distribution. A difference is considered significant if the ranked mean for RNAi::*NbTOL9a* (purple) falls within or beyond the blue percentile of the mean distribution of RNAi::*GUS* (blue) for each treatment combination.
(TIF)

**S15 Fig. Statistical analysis of HR mediated by autoactive NRC3$^{D480V}$, but not MEK2$^{DD}$ or NRC4$^{D478V}$, is reduced when NbTOL9a is overexpressed.** Statistical analysis was conducted using besthr R library (MacLean, 2019) (S11 Data). For (A) MEK2$^{DD}$, (B) NRC3$^{D480V}$, and (C) NRC4$^{D478V}$, the left most panel represents the ranked data (dots) and their corresponding means (dashed lines), with the size of each dot proportional to the number of observations for each specific value (refer to count legend bottom of figure). The right panel shows the distribution of 1,000 bootstrap sample rank means, where the blue areas under the curve illustrate the 0.025 and 0.975 percentiles of the distribution. A difference is considered significant if the ranked mean for AVRcap1b (purple) falls within or beyond the blue percentile of the mean distribution of EV control (blue) for each treatment (MEK2$^{DD}$, NRC3$^{D480V}$, and NRC4$^{D478V}$), in the presence of either EV or NbTOL9a. EV, empty vector; HR, hypersensitive response.
(TIF)

**S16 Fig. Statistical analysis of NRC3$^{D480V}$ cell death suppression mediated by AVRcap1b in NbTOL9a silencing.** Statistical analysis was conducted using besthr R library (MacLean, 2019) (S12 Data). Autoactive NRC3 (NRC3$^{D480V}$) was coexpressed with either EV (negative control) or AVRcap1b (labelled on bottom of plot), with RNAi::*GUS* (blue) or RNAi::*NbTOL9a* (purple) silencing treatments. (A) represents the ranked data (dots) and their corresponding means (dashed lines), with the size of each dot proportional to the number of observations for each specific value (refer to count key, on right hand side). (B) shows the distribution of 1,000 bootstrap sample rank means, where the blue areas under the curve illustrate the 0.025 and 0.975 percentiles of the distribution. A difference is considered significant if the ranked mean for a given condition falls within or beyond the blue percentile of the mean distribution of the corresponding RNAi::*GUS* controls within each given treatment (NRC3$^{D480V}$ + EV, NRC3$^{D480V}$ + AVRcap1b, NRC4$^{D478V}$ + EV, and NRC4$^{D478V}$ + AVRcap1b). EV, empty vector.
(TIF)

**S1 Table. List of effectors used in this study.**
(XLSX)

**S2 Table. List of interactors of AVRcap1b identified in Y2H screen.**
(XLSX)

**S3 Table. List of interactors of SS15 identified in Y2H screen.**
(XLSX)

**S4 Table. List of interactors of AVRcap1b identified by IP-MS.**
(XLSX)

**S5 Table. Sequences of *N. benthamiana* TOLs used in this study.**
(XLSX)

**S6 Table. List of primers used for *P. infestans* effector cloning.**
(XLSX)

**S7 Table. Sequences of NRC2, NRC3, and NRC4 used in this study.**
(XLSX)

**S8 Table. List of primers used for NRC2, NRC3, NRC4, and SW5b cloning.**
(XLSX)

**S9 Table. List of NLRs and corresponding AVRs used in cell death assays.**
(XLSX)

**S10 Table. List of primers used to generate NbTOL9a hairpin silencing construct.**
(XLSX)

**S11 Table. List of primers used for cloning NRC Y2H constructs.**
(XLSX)

**S12 Table. Primers used to generate NbTOL golden gate modules.**
(XLSX)

**S13 Table. List of final OD$_{600}$ used in CoIP experiments.**
(XLSX)

**S14 Table. Spectrum reports for AVRcap1b IP-MS experiments.**
(XLSX)

**S15 Table. Primers used for cloning NRCs into *E. coli* heterologous expression vectors.**
(XLSX)

**S1 Data. HR score data used to generate Fig 1B.**
(XLSX)

**S2 Data. The average HR score of EV versus the average HR score of each tested effector for Pto/AvrPto, for Fig 1B.**
(XLSX)

**S3 Data. The average HR score of EV versus the average HR score of each tested effector for Rpiblb2/AVRblb2, for Fig 1B.**
(XLSX)

**S4 Data. HR score data used to generate Fig 1D, S2 and S4 Figs.**
(XLSX)

**S5 Data. HR score data used to generate Fig 2D, S5 Fig.**
(XLSX)

**S6 Data. HR score data used to generate Fig 3B and 3D, S6 Fig.**
(XLSX)

**S7 Data. HR score data used to generate Fig 4C and 4E, S7 Fig.**
(XLSX)

**S8 Data. Data used to generate Fig 5B and 5C.**
(XLSX)

**S9 Data. Output for statistics for Fig 5B and 5C.**
(XLSX)

**S10 Data. HR score data used to generate Fig 10B, S10 Fig.**
(XLSX)

**S11 Data. HR score data used to generate Fig 11B, S11 Fig.**
(XLSX)

**S12 Data. HR score data used to generate Fig 12B, S12 Fig.**
(XLSX)

**S13 Data. Alignment used to generate phylogenetic tree of *A. thaliana* and *N. benthamiana* TOL proteins.**
(PHY)

## Acknowledgments

We thank Aleksandra "Ola" Białas (The Sainsbury Laboratory, Norwich, UK) for valuable comments on figures for this paper and Adeline Harant for technical assistance.

## Author Contributions

**Conceptualization:** Lida Derevnina, Mauricio P. Contreras, Chih-Hang Wu, Sophien Kamoun.

**Formal analysis:** Lida Derevnina, Jan Skłenar, Dan MacLean.

**Funding acquisition:** Lida Derevnina, Hiroaki Adachi, Jessica Upson, Angel Vergara Cruces, Sophien Kamoun.

**Investigation:** Lida Derevnina, Mauricio P. Contreras, Hiroaki Adachi, Jessica Upson, Angel Vergara Cruces, Rongrong Xie, Abbas Maqbool, Chih-Hang Wu.

**Methodology:** Lida Derevnina, Mauricio P. Contreras, Hiroaki Adachi, Jan Skłenar, Abbas Maqbool, Chih-Hang Wu.

**Project administration:** Lida Derevnina, Mauricio P. Contreras, Hiroaki Adachi, Chih-Hang Wu, Sophien Kamoun.

**Resources:** Lida Derevnina, Hiroaki Adachi, Jessica Upson, Angel Vergara Cruces, Jan Skłenar, Frank L. H. Menke, Sam T. Mugford, Wenbo Ma, Saskia A. Hogenhout, Aska Goverse, Abbas Maqbool, Chih-Hang Wu, Sophien Kamoun.

**Supervision:** Lida Derevnina, Abbas Maqbool, Chih-Hang Wu, Sophien Kamoun.

**Validation:** Lida Derevnina, Mauricio P. Contreras, Jessica Upson, Sam T. Mugford, Abbas Maqbool.

**Visualization:** Lida Derevnina, Jan Skłenar.

**Writing – original draft:** Lida Derevnina, Mauricio P. Contreras, Hiroaki Adachi, Abbas Maqbool, Sophien Kamoun.

**Writing – review & editing:** Lida Derevnina, Mauricio P. Contreras, Hiroaki Adachi, Abbas Maqbool, Sophien Kamoun.

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
