## [Editor Report · Decision Letter 0]

23 Feb 2021

Dear Dr. Kamoun, 

Thank you for submitting your manuscript entitled "Plant pathogens convergently evolved to counteract redundant nodes of an NLR immune receptor network" for consideration as a Research Article by PLOS Biology.

Your manuscript has now been evaluated by the PLOS Biology editorial staff, as well as by an academic editor with relevant expertise, and I am writing to let you know that we would like to send your submission out for external peer review.

Please re-submit your manuscript within two working days, i.e. by Feb 25 2021 11:59PM.

Kind regards,

Paula 

---

Associate Editor

PLOS Biology

---

## [Decision Letter · Decision Letter 1]

31 Mar 2021

Dear Dr. Kamoun,

Thank you very much for submitting your manuscript "Plant pathogens convergently evolved to counteract redundant nodes of an NLR immune receptor network" for consideration as a Research Article at PLOS Biology. Your manuscript has been evaluated by the PLOS Biology editors, an Academic Editor with relevant expertise, and by several independent reviewers.

You will see that both reviewers find the work interesting and well done, but find some issues that will need to be addressed before publication. In particular, both reviewers think that you should discuss the different results you find in the Y2H and IP assays. Reviewer #1 thinks that a greater diversity of NLRs should be tested if they want to claim specificity to this node in the network, including non-NRC-dependent NLRs, says that it’s not clear why NRC2 was omitted, but in his opinion, the importance of the paper does not depend on this, you have to clarify the biological basis to choose only some effectors, in particular only one family of the nematode effectors, says that your data only shows that NbTOL9a is required for NRC3, and is not entirely convinced that the cyst nematodes and P. infestans effectors target the same node. Reviewer #2 wants you to do an experiment to resolve whether TOL9a interacts directly with NRCs to suppress AVRcap1b mediated cell death. Please address the rest of the reviewers' issues. 

In light of the reviews (below), we are pleased to offer you the opportunity to address the comments from the reviewers in a revised version that we anticipate should not take you very long. We will then assess your revised manuscript and your response to the reviewers' comments and we may consult the reviewers again.

We expect to receive your revised manuscript within 1 month.

**IMPORTANT - SUBMITTING YOUR REVISION**

*Resubmission Checklist*

*Published Peer Review*

*PLOS Data Policy*

*Blot and Gel Data Policy*

Sincerely,

Paula

---

Associate Editor,

pjaureguionieva@plos.org,

PLOS Biology

REVIEWS:

Reviewer #1: Nematode effector proteins.

Reviewer #2: Plant immunity and plant-pathogen interactions.

Reviewer #1: This is an extremely important paper for our field.

The importance of this paper is not in the convergent evolution to the same node (indeed I'm not entirely sure about the wording of this - see below) but rather that signalling from an NRC node is suppressed.

In brief: the authors screen a panel of effectors, from a diversity of pathogens, for their abilities to suppress NRC2/3 dependent or NRC4 dependent NLRs. A very small number of effectors were positive in this screen (5), all the data are shown (to be applauded). Experiments on auto active versions of the NRCs suggest two of the five may act directly on the NRCs themselves. Very convincingly, these two were confirmed to suppress a range of NRC2/3 dependent, but not NRC4 dependent, NLRs. The authors then identify interactors of these two effectors in an unbiased Y2H screen. One of which identified several clones corresponding to NRC3 and 4, but not 2. It is extremely encouraging to see these data, although why NRC2 was missing and NRC4 present (and yet in IP we get 2 and 3) could be discussed further. Regardless, no doubt SS15 interacts with NB-ARC domain of at an NRC (in yeast, planta, and vitro). Further experiments suggest SS15 may interact with both inactive and active forms of NRC2 - again these seem robust, NRC3 seems to be generally less amenable to IP in both forms. The other effector, AVRcap1b, also has similar sets of interactors from both Y2H and Co-IP which is extremely encouraging. In this case, it appears to be a protein *downstream* of NRC2/3 involved in membrane trafficking (TOLs). Importantly, it is then shown that a TOL is a negative regulator of auto active NRC3 but not NRC4 nor MEK2 - with the conclusion that it is specific to this node in the network (I think a greater diversity of NLRs should be tested if this claim is wanted to be made, including non-NRC-dependent NLRs). Its not clear why NRC2 was omitted, but again, in my opinion, the importance of the paper does not depend on this. I recommend the manuscript is accepted following minor revisions to the text to either additionally clarify, or rephrase, some of the points raised below.

General comments:

It seems there was a drastic pre-screen of effectors before being tested here (at least for the nematode effectors, only a single family was chosen - omitting most of the diversity of known effectors, including many families involved in immunity suppression) - the biological basis for this choice should be made clear.

It is rather neat that RBP1:GPA2 is suppressed by SS15, because RBP1 is itself a SPRYSEC. Paints a nice picture of one SPRYSEC coming to the rescue of another. Also, rather intriguingly, SS15 seems to be recognised in tabacum - outside the scope of this paper but would be fascinating to know what by, and whether NRC dependent.

Curious that SS10 and 34 also suppress NRC4-mediated Rpi-blb2/AVRblb2 - but apparently not directly (or at least not the auto active versions). The suggestion is made that they must target either the sensor or block the interaction between the sensor and the helper. Struggling a little to understand why they may block the sensor, if that sensor does not recognise a cognate nematode effector - perhaps it points to something in between? Would have been nice to see whether they also suppress R1/AVR1, or what their interactors are, but this is not necessary for this manuscript - perhaps a point for discussion.

The claim that AVRcap1b and SS15 suppress Rx-mediated cell death only in the absence of NRC4 seems to be supported by the data, but it does differ from the literature. This should at least be noted, as it is with HopAB2.

Moffett, Peter, et al. "Analysis of Globodera rostochiensis effectors reveals conserved functions of SPRYSEC proteins in suppressing and eliciting plant immune responses." Frontiers in plant science 6 (2015): 623.

"We found that SS15 directly binds the NB-ARC domain of NRC2 and NRC3, while AVRcap1b associates with NbTOL9a and requires this host protein to fully suppress NRC2 and NRC3."

Data only shown that NbTOL9a is required for NRC3?

"We conclude that cyst nematodes and P. infestans convergently evolved effectors that target the same key nodes of the NRC network to suppress host immune signalling."

I am not entirely convinced that they target the same node, if AVRcap1b is doing something downstream? By the same logic, it could be argued that AVRcap1b also targets the sensor nodes. Without knowing that TOLs are *exclusive* to this node (showing not involved in NRC4 is not sufficient), it seems a little far reaching. Even then, isn't it just the next node in a network? Addressing this could be as simply as "at or downstream of the same node", or "signalling from the same node".

Several comments with the text of Figure 13:

"P. infestans and G. rostochiensis, two evolutionary unrelated pathogens, have evolved effectors that suppress signaling mediated by the NRC network."

I know the intended meaning, but they are not unrelated. Notwithstanding that, I agree with the premise of this sentence as opposed to the above.

"The cyst nematode effector, SS15, can suppress both inactive and activated NRC2 and NRC3 by binding the NB-ARC domain."

I don't think there is any data to show it can suppress the inactive forms, and thinking about it I'm not sure what that means?

"The P. infestans effector, AVRcap1b, can suppress the function of NRC2 and NRC3 by associating with the ESCRT-related protein NbTOL9a."

Only data shown for NRC3 not NRC2?

Minor comments:

Figure 2:

What is the R domain of AVRcap1b and the yellow unlabelled domain of SS15?

Can infiltrated areas be indicated on c as in Figure 1c?

I know it read, "The details of statistical analysis are presented in Figure S4." But I really think at least the name of the test and the cutoff value used should be listed in the main figure and the text. This goes for all figures.

Figure 5:

Its not clear from the diagram how extreme resistance and susceptibility are differentiated.

SPRYSEC is already an acronym. Consider not using an acronym of an acronym and keeping SPRYSEC.

While they are variously used in the literature, I am not sure I agree that parasite is an umbrella term for pests and pathogens - authors may wish to rephrase.

"Parasites as diverse as bacteria, oomycetes, nematodes and aphids turned out to be much more sophisticated manipulators of their host plants than anticipated." 

Were they ever considered unsophisticated? - consider rephrasing.

._To 

"Interestingly, even though AVRcap1b is a robust NTI suppressor, it can activate immunity on accessions of Solanum capsicibaccatum carrying the NLR disease resistance gene Rpi-cap1- hence the moniker AVR (33, 75). The fact that AVRcap1b can act as both a trigger and suppressor of NLR immunity further highlights the complex coevolutionary dynamics that exist between effectors and NLRs and the need for studies that take into account these intricate epistatic interactions."

Same looks to be true for SS15 in N. tabacum.

Reviewer #2: In this paper, the authors screened a library of effectors from different pathogens to identify effectors that suppress sensor NLR-mediated cell death in the Nicotiana benthamiana system. They identified 5 effectors from this screen and tested these effectors to see if they also suppress autoactive helper NLRS, NRC2, NRC3 and NRC4 mediated cell death. From this analysis they identified that AVRcap1b and SS15 effectors suppress NRC2 and NRC3 mediated cell death but not NRC4. 

- The data presented with respect to these experiments are well carried out and conclusions drawn are consistent with the data shown

To identify proteins that interact with these two effectors they performed Y2H screen. The co-IP data presented shows that SS15 effector interacts with NRC2 and NRC3 but not with NRC4.

- In the Y2H experiment, the authors recovered NRC3 and NRC4 with SS15. However, in the co-IP experiments it only interacted with NRC3. In co-IP the authors show that SS15 also interacts with NRC2 but this was not detected in Y2H screen. Some discussion should be included with respect to the different assays showing different results.

The data shown indicate that SS15 interacts directly with the NB-ARC domain of NRC4 in vitro. Furthermore, it interacts with both activated and inactivated NRC2 in planta. 

To understand how AVRcap1b suppresses cell death, the authors performed pulled experiments and identified TOL family protein. Using co-IP, transient overexpression and silencing experiments, the data shown indicate that AVRcap1b may use TOL9a host factor to suppress NRC-mediated cell death.

- Does TOL9a interact directly with NRCs to suppress AVRcap1b mediated cell death? 

The manuscript is well written and easy to follow. The data shown indicate that considering NRC network is important for immunity, the effectors target this branch of the immune network to suppress cell death response of immunity.

---

## [Editor Report · Decision Letter 2]

14 Jul 2021

Dear Dr. Kamoun,

Thank you for submitting your revised Research Article entitled "Plant pathogens convergently evolved to counteract redundant nodes of an NLR immune receptor network" for publication in PLOS Biology. I have now discussed your revision with the Academic Editor. 

We will probably accept this manuscript for publication, provided you satisfactorily address the following data and other policy-related requests.

DATA POLICY:

Regardless of the method selected, please ensure that you provide the individual numerical values that underlie the summary data displayed in the following figure panels as they are essential for readers to assess your analysis and to reproduce it: Figures 1BD, 2D, 3BD, 4CE, 5BC, 10B, 11B, 12B, S2ABCD, S4ABCDEF, S5ABCDEF, S6ABCD, S7ABCDEF, S14ABCD, S15ABC, S16AB.

**Please also ensure that figure legends in your manuscript include information on where the underlying data can be found, and ensure your supplemental data file/s has a legend.**

We expect to receive your revised manuscript within two weeks.

*Published Peer Review History*

*Early Version*

Sincerely,

Paula

---

Associate Editor,

pjaureguionieva@plos.org,

PLOS Biology

---

## [Editor Report · Decision Letter 3]

27 Jul 2021

Dear Dr. Kamoun,

On behalf of my colleagues and the Academic Editor, Xinnian Dong, I am pleased to say that we can in principle offer to publish your Research Article "Plant pathogens convergently evolved to counteract redundant nodes of an NLR immune receptor network" in PLOS Biology, provided you address any remaining formatting and reporting issues. These will be detailed in an email that will follow this letter and that you will usually receive within 2-3 business days, during which time no action is required from you. Please note that we will not be able to formally accept your manuscript and schedule it for publication until you have made the required changes.

PRESS

Sincerely, 

Paula 

---

Paula Jauregui, PhD 

Associate Editor 

PLOS Biology
